# Pipelined and conflict-free number theoretic transform accelerator for CRYSTALS-Kyber on FPGA

Ayesha Waris *, Arshad Aziz, Bilal Muhammad Khan

National University of Sciences and Technology (NUST), Karachi, Pakistan

* ayesha.waris@pnec.nust.edu.pk

## Abstract

Post-quantum cryptographic (PQC) algorithms are essential due to the threat posed by quantum computers to the security of currently deployed cryptosystems. CRYSTALS-Kyber, based on Lattice-based cryptography, has been standardized as the Public-Key Encryption and Key-Establishment Mechanism Algorithm by the National Institute of Standards and Technology (NIST). An efficient hardware acceleration of CRYSTALS-Kyber relies on optimizing the computationally intensive Polynomial Multiplication Number Theoretic Transform (NTT) unit. This work presents an FPGA implementation of conflict-free and pipelined single-path delay feedback based NTT core for Kyber by employing various architectural optimizations including pipelining, resource sharing and algorithmic optimizations like multiplier-less Montgomery reduction algorithm. As a result, our design has 7.8% reduction in resources and 49.6% improved Area-Time Product (ATP) as compared to the state-of-the-art designs. The presented architectures are coded using Verilog HDL and implemented on Xilinx Artix-7 XC7A100T-3 and Virtex-7 XC7VX485T-3 devices using Vivado Design Suite 2022.2.

## 1. Introduction

Quantum Computing is a fast-emerging technology that exploits properties of Quantum mechanics to perform calculations much faster than our conventional computers. While the high computation capability of quantum computers is extremely advantageous, it also poses a serious threat to the security of currently deployed public-key cryptosystems such as RSA [1], Elliptic Curve Cryptography [2]. Consequently, to protect the confidentiality of classified information it is significant to develop quantum-resistant cryptosystems.

In 2016, the National Institute of Standards and Technology (NIST) initiated a Post Quantum Cryptography (PQC) competition [3] to standardize cryptographic algorithms that can resist quantum attacks and substantial research has been done

**Data availability statement:** The author generated code is available at Figshare under the following DOI 10.6084/m9.figshare.30371407. https://figshare.com/articles/software/Number_Theoretic_Transform_for_CRYSTALS-Kyber/30371407.

**Funding:** The author(s) received no specific funding for this work.

**Competing interests:** No authors have competing interest.

in this field since then. After a six year competition NIST declared CRYSTALS-Kyber [4] as a standard in the Public-Key Encryption (PKE) and Key Encapsulation Mechanism (KEM) category in July 2022. CRYSTALS-Kyber is a Lattice-based cryptographic (LBC) scheme [5], and its security is based on the hardness of solving the Module-Learning with errors (M-LWE) problem. Also known as Kyber, this KEM provides good trade-off between security, performance, message sizes and flexibility as compared to its counterparts in the NIST competition.

Polynomial multiplication [6,7] is the most computationally expensive operation in LBC schemes. Considerable research has been done to reduce the complexity of polynomial multipliers by proposing several multipliers: School-book, Karatsuba-Ofman, Toom Cook and NTT. The last one, known as the Number Theoretic Transform (NTT) [8] is a variant of Fast Fourier Transform (FFT) and reduces the number of multiplication complexity from $O\left(n^2\right)$ to $O\left(nlogn\right)$ and is used by Kyber for polynomial multiplications. Hence, to make Kyber efficient, acceleration of NTT is imperative.

To reduce the performance bottleneck, NTT has been implemented and investigated on various platforms. While software implementations [9] provide programming flexibility, hardware platforms such as ASICs and FPGAs [10–12], can be accelerated using optimization strategies such as parallelization and resource sharing. Hardware/Software co-design platforms are also very popular and implement a software-based processor and hardware based design on a single chip. Reconfigurability and flexibility provided by FPGAs make them an ideal choice for implementing computationally intensive algorithms such as NTT. However, providing an optimal trade-off between resource utilization, processing delay and energy consumption is the major challenge for the hardware designers.

In this work, we have proposed various hardware and algorithmic optimizations and designed an FPGA based NTT polynomial multiplier to accelerate performance of CRYSTALS-Kyber. Research Flow, scheme and methodology of this work is presented in Fig 1 and block diagram of overall architecture is given in Fig 2.

Main contributions are as follows:

- This work first implements conventional single-path delay feedback-based NTT (SDFNTT) architecture for Kyber on FPGA. The paper highlights that incorporating pipelining to increase the maximum frequency of the SDFNTT leads to data conflicts. A novel pipelined SDF-based NTT architecture is then proposed that resolves the data conflicts by introducing an alternative data flow.

- By using only a small additional hardware resource in each stage, our pipelined SDFNTT achieves a significantly higher clock frequency of approximately 40% on the Artix-7 and 32% on the Virtex-7 device, compared to the non-pipelined SDFNTT implementation.

- The pipelined SDFNTT incorporates 14 pipeline stages. To further increase the pipelining, configurable FIFO depths are presented, allowing the depth to be adjusted according to the number of pipeline stages to avoid data conflict.

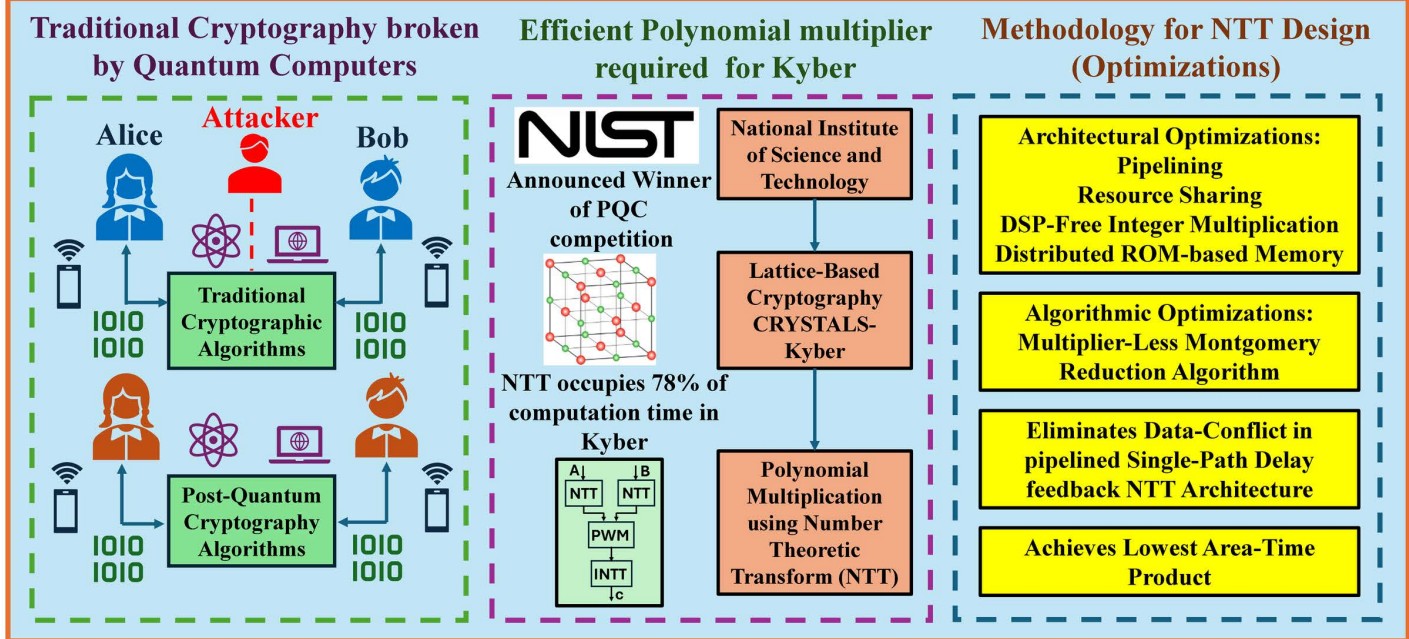

**Fig 1. Schematic and sequence of the overall Research procedure for Pipelined Number Theoretic Transform Accelerator for CRYSTALS-Kyber on FPGA.**

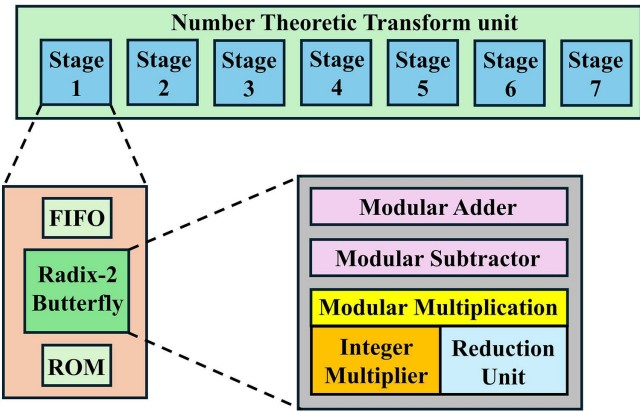

**Fig 2. Block diagram of the proposed overall architecture.**

- Multiple architectural optimizations such as multiplier-less modular multiplier, unified butterfly unit, distributed ROM based memories for twiddle factors storage are adopted to minimize resource consumption of the proposed design and resulted in achieving 49.6% improved ATP when compared to the prior works.

  The paper is structured as: literature review is elaborated in section 2, preliminaries are discussed in section 3, section 4 presents the proposed SDFNTT architecture, followed by results discussions and comparisons in section 5.

## 2. Literature review

Hardware implementations of CRYSTALS-Kyber mainly focusses on NTT as it is the most computationally intensive and resource-hungry part of the design. The complex memory access pattern and large memory requirement is the bottleneck for accelerating the performance of the NTT unit. There has been various research on NTT for Fully Homomorphic Encryption (FHE) and Ring Learning with Errors (RLWE) schemes, but in this paper, we have limited our discussion to the NTT unit of PQC schemes in NIST competition mainly Kyber.

NTT unit is implemented using software, hardware/software co-design or pure hardware approaches. Abdulrahman et al. [9] optimized and implemented CRYSTALS-Kyber KEM and Dilithium signature schemes on Cortex-M4. Authors in [13] implemented three PQC algorithms: FrodoKEM, NewHope and Kyber, on a GPU to accelerate their operations. Botros et al. in [14], proposed a Cortex-4 memory-efficient implementation of Kyber. An ARMv8 microprocessor-based optimized implementation was presented by Nguyen et al. for Kyber, NTRU, and Saber in [15]. Interested readers can find the survey of software implementations of NTT (ARM Cortex, GPU) in [16]. Nguyen et al. in [17] and Dang et al. in [18] used hardware/Software co-design approach and introduced an HLS-based NTT architecture for various round 2 PQC algorithms. The execution time for encapsulation and decapsulation was improved in these studies, compared to the pure software designs.

Inherent parallelism [19] in NTT structure makes pure hardware the best choice for its implementation. In recent years, NTT has been widely implemented on hardware platforms such as ASICs and FPGAs. In [20], Bisheh et al. have provided an instruction set architecture of Kyber and implemented it on ASIC. In this architecture, the execution time of the NTT is reduced by merging pre-processing into the NTT algorithm. Banerjee et al. in [21] have proposed a power efficient RISC-V architecture based crypto-processor on ASIC for several PQC algorithms such as Frodo, qTesla, Kyber, Dilithium and NewHope. Due to its reconfigurability and flexibility, FPGA is a popular platform for implementation testing and result validation for PQC, FHE and RLWE algorithms. According to literature, FPGA implementations of Kyber's can be classified into two categories, iterative and pipelined. Table 1 summarizes techniques adopted by a few previous works implementing Kyber's NTT core on FPGA and details are presented in the ensuing paragraphs.

Huang et. al [30] proposed iterative CT/GS butterfly for NTT/INTT computation which heavily relies on Block RAM units. In [19], Authors have implemented an iterative NTT with CT and GS based eight butterfly units which reduces the NTT

**Table 1. Techniques adopted in former works for implementing NTT/INNT Kyber core.**

| S. No | Author name/Year | Techniques used |
|---|---|---|
| 1 | Sun et al. [22] (2024) | Radix-4 based NTT design (Iterative) |
| 2 | Nguyen et al. [23] (2024) | Radix-2 and radix-4 based NTT unified architectures and Dilithium. (Pipelined) |
| 3 | Guo et al. [24] (2023) | Mixed radix 2/4 NTT along with memory mapping and access scheme. (Iterative) |
| 4 | Ni et al. [25] (2023) | Radix-2 MDC-based NTT. (Pipelined) |
| 5 | Itabashi et al. [26] (2022) | Radix-2, 2-parallel radix-2 and radix-4 NTT architectures. (Iterative) |
| 6 | Ye et al. [27] (2022) | NTT/INTT core with flexibility in input parameters. (Pipelined) |
| 7 | Bisheh et al. [28] (2021) | K-RED and KRED-2x reduction unit along with a reconfigurable radix-2 butterfly core. (Iterative) |
| 8 | Zhang et al. [29] (2021) | Hardware efficient NTT with ping-pong memory access scheme. (Iterative) |

iteration time. Chen et al. [31] designed an optimized NTT by modifying the CT/GS butterfly and modular arithmetic units. Zhang et al. have proposed a ping pong memory access scheme and a fast modular reduction based on bitwise add shift operations along with look-up table in [29]. Yaman et al. [32] have implemented a unified butterfly structure and three different architectures (lightweight, balanced, high-performance) for Kyber-specific NTT, using 1, 4 and 16 butterfly units, respectively. A $2 \times 2$ butterfly unit based NTT architecture along with K2RED algorithm is presented in [28] which improves the computation time. Xing et al. [33] have implemented a Kyber processor on the Artix-7 platform. The architecture used two butterfly units to process the even and odd input coefficients. In [34], Gao et al. proposed a conflict-free memory access pattern and performed NTT, INTT and PWM calculations. Recently, Gao et al. presented a memory mapping and access scheme for Kyber-optimized NTT and PWM, together with a mixed radix-2/4 approach [24]. Itabashi et al. have proposed three NTT architectures, radix-2, 2-parallel radix-2 and radix-4, of Kyber in [26]. An optimized version of modular reduction Exact-KRED for Kyber NTT is proposed in [11], along with a $2 \times 2$ butterfly unit NTT structure. To make the memory management compact and simpler, in [35] Ni et al. have proposed an iterative NTT/INTT architecture, with point-wise multiplication and have replaced all the BRAM units with three FIFOs. Sun et al. in [22] designed a radix-4 NTT/INTT core along with a conflict-free memory access scheme.

Pipelined NTT architectures are gaining popularity among researchers because memory management in these architectures is less complex compared to the iterative approach. In [36], Nguyen et al. have introduced a dual-path delay feedback (DDF) based NTT for Kyber. The input coefficients of NTT are divided into two parts for processing, which increases the execution time of NTT. The pipelined architecture is proposed for NTT for various schemes with different parameters [27]. Kyber parameters $n = 256, q = 3329$ are also mentioned in the paper. Ni et al. [25] have presented a radix-2 based multi-path delay commutator (MDC) NTT/INTT architecture using resource sharing. In [23] Nguyen et al. have adopted MDC approach and proposed radix-2 and radix-4 unified NTT architectures for Kyber and Dilithium. Two MDC-based NTT designs are proposed in [37]. The unified 2-parallel and 4-parallel architectures are reconfigurable and support NTT, INTT and PWM operations.

## 3. Preliminaries

### 3.1. Classical NTT versus NTT in Kyber

NTT is a variant of Fast Fourier Transform (FFT) in the finite field. The complexity of NTT multiplication has decreased from $O\left(n^2\right)$ to $O\left(nlogn\right)$ [8], when compared to the traditional School-book algorithm. NTT based polynomial multiplication can be represented by $c = INTT\left\{NTT\left(a\right) \circ NTT\left(b\right)\right\}$, where $a$, $b$ are the input polynomials and $c$ is the output polynomial. Inputs $a$ and $b$ are first transformed into the NTT domain followed by a point-wise multiplication operation (PWM represented by $\circ$). To complete the multiplication and transform back the result into the original domain, inverse NTT (INTT) is applied to the result of PWM.

NTT transformation [38] is performed on polynomial quotient ring $R_q = \mathbb{Z}_q[x]/\left(x^n + 1\right)$, where $\mathbb{Z}_q$ is the ring of integers modulo $q$, $x^n + 1$ is the irreducible polynomial, $n$ is the power of 2 and $q$ is the prime modulus satisfying $q \equiv 1(\mod n)$. For an $n$-point input polynomial $a$, output polynomial in NTT domain is given by $A[i] = \sum_{j=0}^{n-1} a[j].w^{ij} \, odq$ where $0 \leq i < n$. Primitive $n$-th root of unity $w$, is the smallest integer in the ring that satisfies $w^n \equiv 1\,(\mod q)$. INTT can be calculated by multiplying the scaling factor $n^{-1}$ and using negative powers of $w$ and is represented as $a[i] = n^{-1} \sum_{j=0}^{n-1} A[j].w^{-ij} \, odq$.

NTT transformation is performed by appending zeros at the end of input polynomials expanding their degree from $n$ to $2n$. This results in an increase in resources as well as computation time. Negative-wrapped Convolution (NWC), optimization requires $n$-point input polynomials and can only be applied when $2n$-th root of unity, $\psi$ exists, where $\psi^2 \equiv w \mod q$ and $q \equiv 1(\mod 2n)$. Input polynomials in NTT operation are multiplied by powers of $\psi$ and output polynomials are multiplied by negative powers $\psi^{-1}$, known as pre-processing and post-processing steps, respectively. The NWC based NTT and INTT can now be represented as $A[i] = \sum_{j=0}^{n-1} a[j].\psi^j.w^{ij} \, odq$ and $a[i] = n^{-1} \sum_{j=0}^{n-1} A[j].\psi^{-i}.w^{-ij} \, odq$ [38].

Decimation in Time (DIT) and Decimation in Frequency (DIF) are two approaches to implement NTT, which implements Cooley-Tuckey (CT) [39] or Gentleman-Sande (GS) [40] butterfly structures respectively. Both CT and GS butterflies take two inputs and give two outputs, but the output differs because of the placement of multiplier. The output of CT butterfly is $A+B.w$, $A-B.w$ and GS butterfly is $A+B$, $(A-B).w$, where $A$ and $B$ are input coefficients and $w$ is the twiddle factor. Structures of radix-2 CT and GS butterfly are shown in Fig 3a and 3b.

CRYSTALS-Kyber uses modulus $q=3329$ and $n=256$, which is not NWC-NTT friendly as it does not satisfy $q \equiv 1 (\mathrm{mod} 2n)$. Calculating the NTT of Kyber requires a special trick called Truncated-NTT and can be achieved by employing an incomplete FFT trick or by splitting the polynomial ring as mentioned in [38]. By splitting the polynomial ring in Kyber we divide the NTT calculation into even and odd parts [41]. As $n=128$ and $q=3329$, parameters now fulfill the NWC-NTT condition and NTT can now be applied to two smaller polynomial rings.

In this paper, we have adopted the incomplete FFT trick which requires the last stage in Kyber-NTT to be cropped. The output of NTT will now be $1$-degree polynomials, unlike linear terms obtained while using full NTT. Point-wise multiplication operation in Kyber is polynomial multiplication of two $1$-degree polynomials which consists of five multiplications and two additions, different from the original PWM operation which only requires coefficient-wise multiplication of the linear terms obtained after full NTT.

### 3.2. Iterative and pipelined architectures

Like FFT, NTT can be designed using iterative or pipelined architectures. Iterative NTT in its simplest form incorporate a butterfly unit and a memory unit. NTT is calculated by loading the data from the memory into the butterfly unit and then storing the outputs back into the memory. This is done iteratively till butterfly operations in all the stages of the NTT are computed. Different iterative implementations of Kyber are reported in the literature [11,19,20,22,24,26,28–35], using single or multiple butterfly and memory units and using optimized data flow and memory access techniques. These architectures suffer from high memory utilization and high implementation complexity.

Unlike iterative designs, pipelined architectures process data in a continuous flow and critical paths of the architecture can be removed by adding registers which results in high frequency while using low-area resources. An n-point pipelined NTT architecture consists of $log_2 n$ stages where each stage has one butterfly unit and performs $n/2$ butterfly operations. A few configurations of Pipelined architectures of Kyber are available in literature [23,25,27,36], Single-path delay feedback (SDF) and Multi-path delay-commutator (MDC) being the most popular ones.

### 4. Proposed architecture

Polynomial multiplication via NTT units are the most significant, complex and time-consuming operations for computation of Kyber as they occupy $78$% of computation time [28]. To improve the performance of Kyber, an efficient NTT core is required which provides a high throughput and uses low hardware resources.

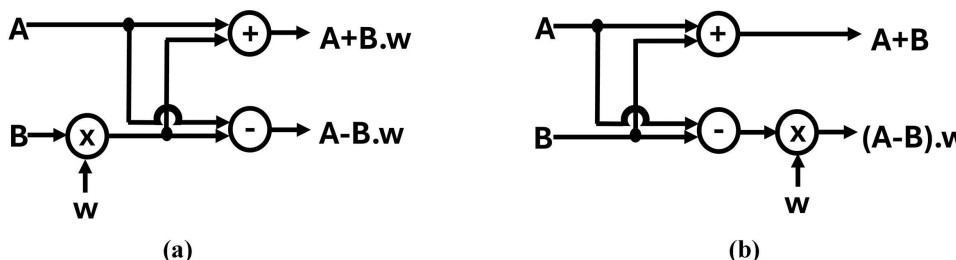

**Fig 3. Radix-2 butterfly structure.** (a) Cooley Tuckey (b) Gentlemen Sande.

In this work, we have implemented two SDF-based NTT architectures, non-pipelined and pipelined, for CRYSTALS-Kyber. The conventional single-path delay feedback (SDF) architecture is a simple, non-pipelined, BRAM-free design with a straightforward memory access pattern that continuously streams input data and performs NTT computations across successive stages. However, its maximum operating frequency is limited because introducing pipelining leads to data conflicts. This work presents a novel pipelined SDF-based NTT architecture that addresses these data conflict issues by introducing an alternative data flow and varying the depth of integrated FIFOs. As a result, the proposed design achieves significantly higher frequency compared to the conventional SDF-based NTT. Furthermore, to enhance pipelining flexibility, configurable FIFO depths are provided for each stage.

High performance of NTT is achieved using various optimizations on architectural and algorithmic level. We have accelerated NTT by employing techniques such as pipelining, using LUT-based Multiplication for integer multiplication unit, LUT-based FIFOs for data buffering, utilizing distributed ROM-based memories, replacing constant multiplications with lightweight operations in modular reduction unit and designing a reconfigurable butterfly core using resource sharing. As NTT and INTT operations are not performed simultaneously, we have used the same architecture to perform both operations. Different data paths have been provided to alternate the data flow between NTT and INTT computations.

In the first few sections low-level arithmetic modules such as modular adder, modular subtractor, modular multiplier and unified butterfly structure are discussed. Then our non-pipelined SDFNTT, which is a conventional SDF architecture, along with its main components and dataflow is explained. Subsequently, issues arising by adopting pipelining in SDF architectures are elaborated and then the proposed pipelined SDFNTT is presented which resolves the data conflict issues in the formerly implemented architecture. At the end, we discussed our unified NTT/INTT SDF architecture.

### 4.1. Modular addition and subtraction

Modular adder and Modular subtractor, presented in Fig 4a and 4b, are optimized for CRYSTALS-Kyber, using Kyber's Modulus $q = 3329$. The bit *bor_car* selects the output of the multiplexer in both units.

### 4.2. Modular multiplier

An efficient and resource-saving modular multiplication unit is significant in designing a compact and fast NTT. Our modular multiplier is divided into two parts (a) Integer multiplier and (b) Montgomery reduction unit.

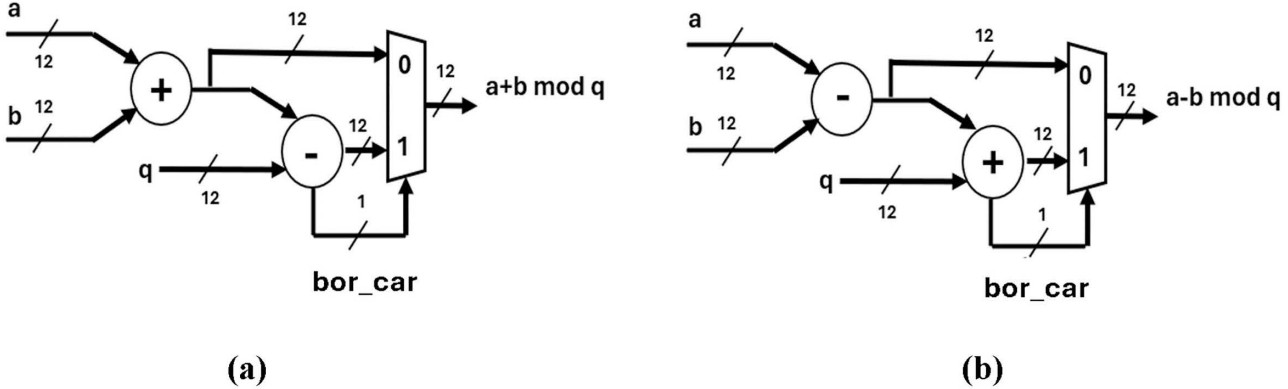

**(a)** **(b)**

**Fig 4. Arithmetic modules.** (a) Modular adder (b) Modular subtractor.

**4.2.1. Integer multiplier.** The integer Multiplier unit in modular multiplication in Kyber takes two inputs $A$ and $B$ of $12$ bits and gives $24$ bits output. As coefficients have a length of $12$ bits, using DSP48E1 block in $7$-series Xilinx FPGA that supports $25 \times 18$ multiplication [42], will result in partial usage of resources. Our SDFNTT has seven stages of butterfly units, using seven DSP48E1 block-based modular multiplication units will result in a considerable increase in hardware resources.

We have adopted shift and add [43] technique and modified it to design our integer multiplier. Our experiment and results concluded that the proposed $6 \times 6$ multiplier shown in Fig 5 is resource-efficient than other multiplier architectures (Wallace, Booth, Dadda). The presented multiplier takes two $6$ bit input and results in a $12$ bit output. Six bitwise AND operations are performed between six bits of multiplicand $A$ and each bit of Multiplier $B$. The generated partial products are then added using a chain of parallel ripple carry adders. We have reduced the critical path delay by exploiting parallelism in FPGAs and have implemented four units of 6x6 multiplier units $\{A_H[11:6], B_H[11:6]\}$, $\{A_L[5:0], B_H[11:6]\}$, $\{A_H[11:6], B_L[5:0]\}$, $\{A_L[5:0], B_L[5:0]\}$ The output of each multiplier unit is then accumulated using the Vedic tree adder, as shown in Fig 6.

**4.2.2. Modular reduction unit.** In literature, there are two main approaches for implementing modular reduction: Barrett [44] and Montgomery Reduction [45]. The classical Barrett and Montgomery algorithms avoid costly division operations by replacing them with multiplication and shifting operations. Over the period of years, different variations [46,47] of both algorithms have been proposed to accelerate LBC and PQC.

Barrett reduction has been widely employed for designing reduction units in Kyber [19,20,33,48]. SAMS2 is a variation of the Barrett algorithm [47], in which simple inexpensive operations such as bit shifts, addition and subtraction are used instead of constant multiplications. It has been applied in [49,50] for a specific modulus. In [51] Longa et al proposed K-RED based reduction technique which is based on the characteristics of Proth prime numbers of the format $q = k.2^m + 1$ and has contributed to the implementation of efficient reduction units for Kyber in [11,28]. Reduction units in [32,34,52], utilizes the property $2^{12} \equiv 2^9 + 2^8 - 1 \pmod{3329}$ recursively, till the higher-order bits are reduced into lower-ordered bits. The partial results are then added using a chain of adders.

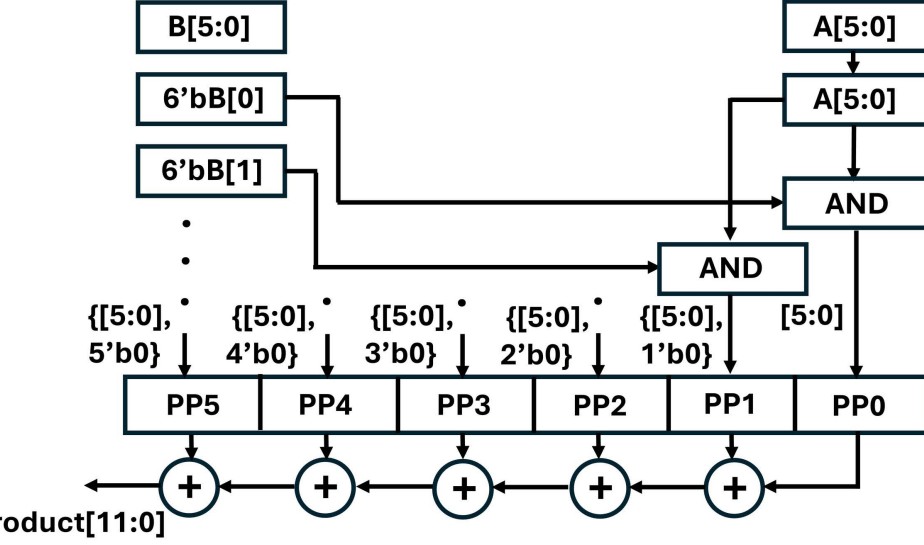

**Fig 5. 6x6 bit integer multiplication unit.**

```
Algorithm 1. Montgomery reduction algorithm, pre-computed Montgomery constant μ Input: q, P such
that 0 ≤ P < q²
```

**Output:** $C \equiv PR^{-1} \bmod q$ such that $0 \leq C < q$
```
  1:  C1 = μ(P mod R) mod R
  2:  C = (P + C1.q)/R
  3:  if (C ≥ q) then
  4:  C = C - q
  5:  end if
  6:  return C
```

For our NTT design, a custom-built reduction unit has been implemented (Fig 7) for Kyber based on the Montgomery reduction algorithm. All the multiplications in the traditional Montgomery algorithm (Algorithm 1) are replaced with lightweight and inexpensive shifting, addition and subtraction operations. Optimized Montgomery algorithm is presented in Algorithm 2. A dedicated modular reduction unit is designed with Kyber's modulus q = 3329 and by utilizing the characteristic property of this prime number. We have adopted the approach in [21], in which modulus $q$ is written in the form $2^{l1} \pm 2^{l2} \pm \cdots \pm 1$, and the constant multiplications in the Barrett algorithm are replaced by bit shifts, addition and subtraction operations.

The input to our Montgomery reduction unit is $24$ bits $P$. The module implements $C = P \bmod q$, using bit shifts, subtraction and addition operations and calculates a $12$-bit output. $R$ is selected to be $4096 = 2^{12}$, where $R > q$. By selecting $R$ as a power of $2$, all the mod $R$ operations in Algorithm 1 are replaced by hardware friendly operations. In step 1 Algorithm 1,

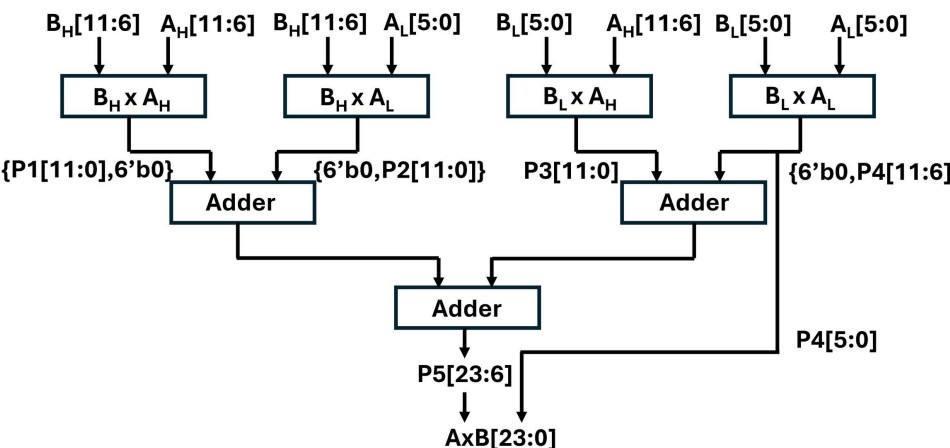

**Fig 6. 12x12 bit integer multiplication unit.**

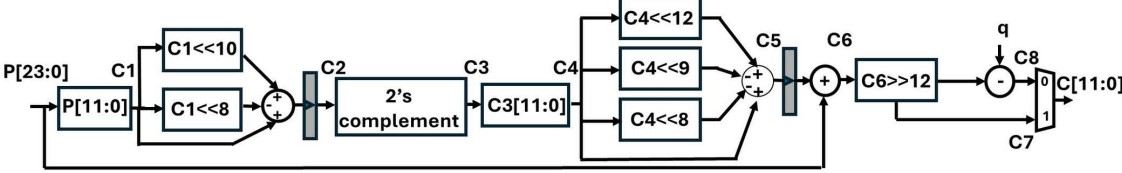

**Fig 7. Montgomery reduction unit.**

the input $P$ is $24$ bits and $P \bmod R$ is the selection of the rightmost $12$ bits of $P$, that is $P[11:0]$. In step 2, division by $R$ is replaced by shifting the bits of the dividend to the right by a factor of $12$. All these algorithmic optimizations are mentioned in Algorithm 2.

```
Algorithm 2. Optimized Montgomery reduction for CRYSTALS-Kyber.
Input:  q, P such that 0 ≤ P < q²
Output: C ≡ PR⁻¹mod q such that 0 ≤ C < q
  1:  C1 = P[11 : 0]
  2:  C2 = C1 << 10 – C1 << 8 + C1
  3:  C3 = Two's complement of C2
  4:  C4 = C3[11 : 0]
  5:  C5 = C4 << 12 – C4 << 9 – C4 << 8 + C4
  6:  C6 = C5 + P
  7:  C7 = C6 >> 12
  8:  C8 = C7 – q
  9:  if (C8 < 0) then
 10:    C = C7
 11:  else
 12:    C = C8
 13:  end if
 14:  return C
```

We have pre-computed Montgomery constant $\mu = -769 \bmod R$ using (1). $769$ is a Solinas Prime. These generalized Mersenne Primes were suggested by Solinas in [53]. Using Montgomery friendly primes help in accelerating Montgomery Multiplications on software and hardware platforms [21,54].

$$\mu = -q^{-1} \bmod R \qquad (1)$$

We have two constant multiplications in Algorithm 1, in step 1 multiplication by $\mu$ and in step 2 multiplication by modulus $q$. These two primes can be represented as $3329 = 2^{12} - 2^9 - 2^8 + 1$ and $769 = 2^{10} - 2^8 + 1$. By using this technique, the constant multiplications are replaced by left bit shifts, additions and subtractions as shown in step 2 and step 5 in Algorithm 2. We have taken the 2s' complement in step 3 to incorporate the negative sign in our calculated μ.

The drawback in Montgomery multiplication is that the calculations are done in the Montgomery domain, it means pre and post-Montgomery calculations are required. The output of Algorithm 2 is $C = PR^{-1} \bmod q$, it implies the result must be multiplied with $R$ to get $C = P \bmod q$ and this constant multiplication will require more hardware resources. One of the inputs to be multiplied by the integer multiplier is the precomputed twiddle factor $w$ which is stored in the memory. In our design, we have multiplied $w$ with $R$ and stored $w.R$ in the memory instead of $w$. This pre-computation will result in $C = P \bmod q$ at the output of our reduction unit.

### 4.3. Unified butterfly unit

NTT-based polynomial multiplier requires input and output coefficients to be in natural order and implement both, forward NTT and inverse NTT transformations. DIT-based NTT transformation takes input in natural order and gives output in bit-reversed order while DIF-based NTT does the opposite. Using only one approach for both NTT and INTT will result in an additional cost of implementing a bit-reversal unit. Different configurations to implement NTT and INTT are given in [55].

In this design, the bit-reversal operation has been avoided, by designing a unified butterfly architecture for NTT/INTT. DIT-based NTT receives inputs in the natural order and results in bit-reversed output. This output will be given as input to the DIF-based INTT which takes bit-reversed input coefficients and results in natural order output. The data flow for NTT and INTT is illustrated in Figs 8 and 9. **a0, a1, . . . ., a7** are normal order coefficients and **a′0, a′1, . . .a′7** are bit-reversed order coefficients.

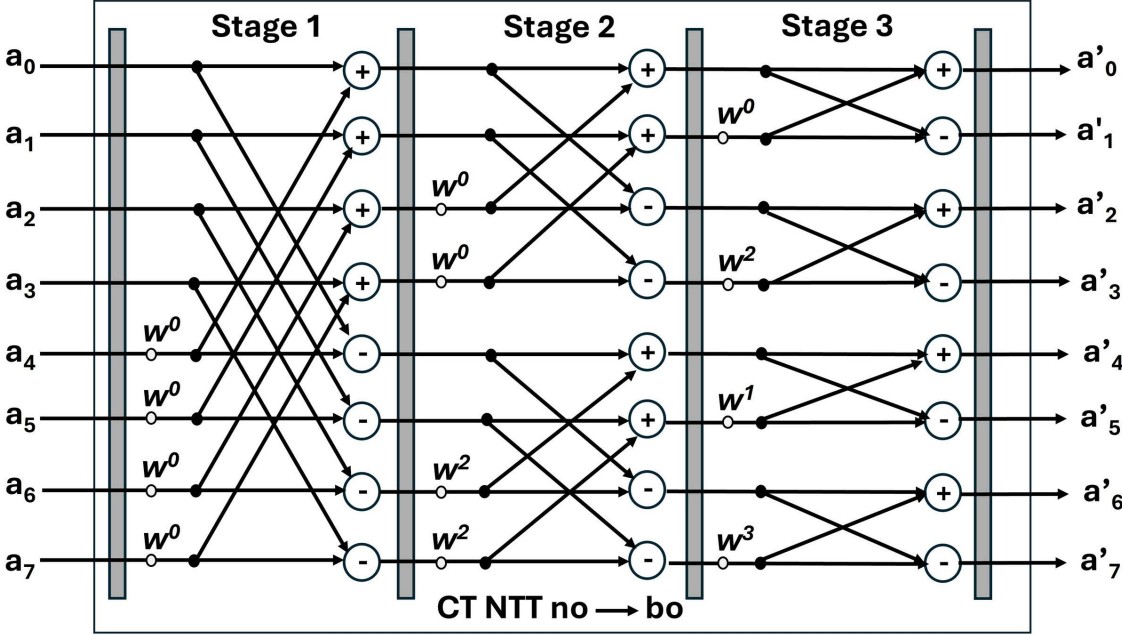

**Fig 8. 8-point DIT NTT data flow.**

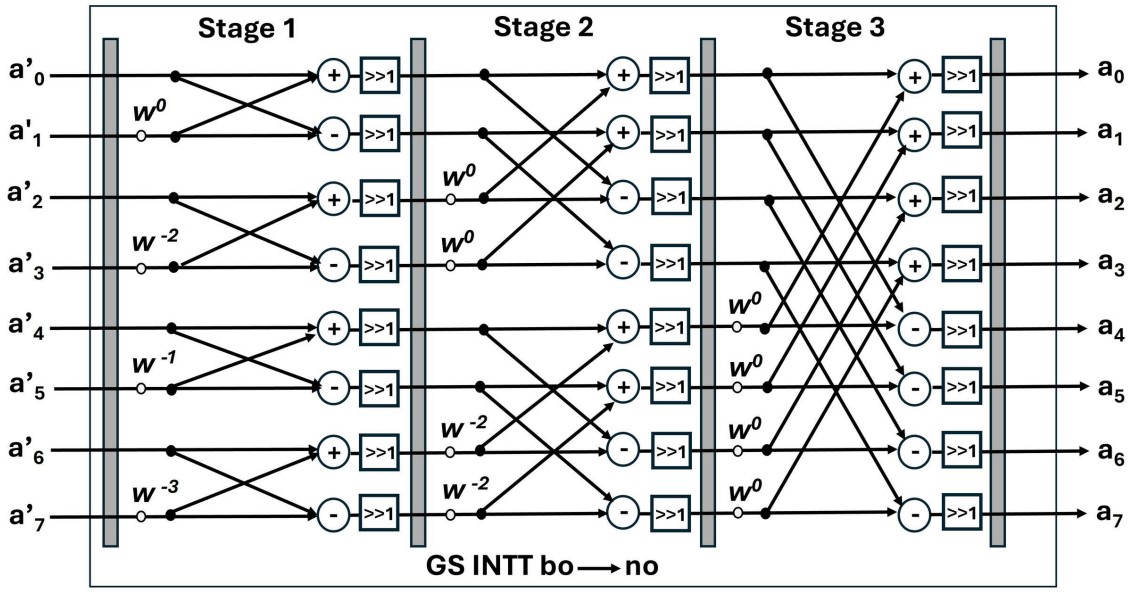

**Fig 9. 8-point DIF NTT dataflow.**

Presented butterfly unit (BU) takes three 12-bit input coefficients *A, B* and *w* results in two output coefficients *Bo*1 and *Bo*2. BU comprises of one modular adder, one modular subtractor and one modular multiplier. The *Mode* control signal given to muxes in the design (Fig 10), selects CT-based butterfly structure for NTT if *Mode* is 1, and GS-based butterfly for INTT operation, if *Mode* signal is 0.

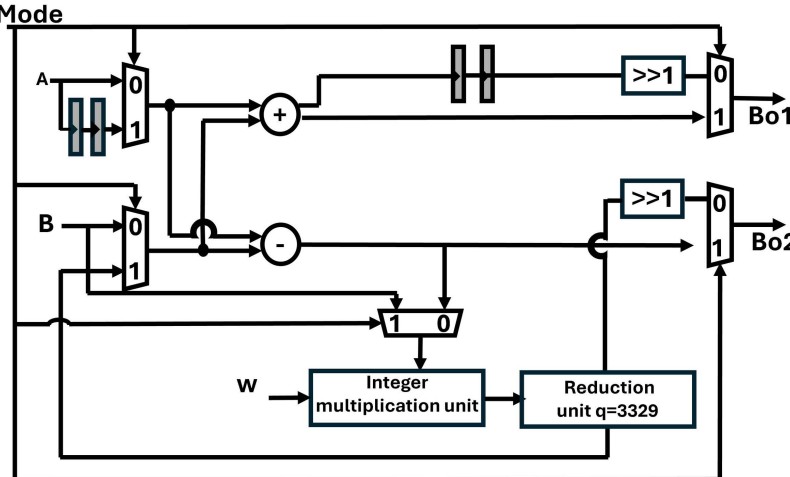

**Fig 10. Unified CT/GS radix-2 butterfly architecture.**

The reduction unit becomes the critical path of the whole architecture due to the LUT-based shift and addition operations. To reduce the critical path and obtain better performance, we have adopted pipelining and added a two-level pipeline in the Montgomery reduction unit. Pipeline registers are also added in our reconfigurable butterfly architecture to synchronize the output coefficients.

If $R_p$ pipeline registers are increased in the reduction unit then for balancing the timing of the output coefficients, $R_p$ number of registers are also added in the input path of $A$ in CT mode and in $A + B$ path in GS mode. In CT mode of our butterfly unit, modular multiplication of $B$ and $w$ takes two cycles. We have inserted two pipeline stages in the input path of $A$ so that both $A$ and $B.w$ are synchronized for modular addition and modular subtraction. In the GS configuration, two pipeline registers are added in the $A + B$ path as multiplication of $A - B$ with $w$ will have a delay of two cycles. Hence, our fully pipelined BU requires a latency of two clock cycles (cc) whether working in a CT or GS mode.

For INTT operation a final scaling by $n^{-1}$ is required at the output of GS butterfly. To implement it, Zhang et al in [56] proposed to insert "$1/2$" operation to obtain $(A + B)/2$ and $\{(A - B).w\}/2$ at the output. This can be easily achieved by shifting in hardware instead of using extra resources for multiplication with $n^{-1}$. In our unified BU, we have inserted a $1$-bit right shift block ">> $1$" to incorporate the INTT operation.

### 4.4. Non-pipelined SDFNTT

**4.4.1. Dataflow in SDF unit.** In our first design, the SDF unit in Fig 11, consists of a radix-2 butterfly unit designed in section 4.3 (excluding pipelining), a FIFO unit for data buffering in the feedback with the BU and a TWROM for storing twiddle factors. The SDF unit will accept one input coefficient and will result in one output coefficient per clock cycle.

Fig 12 represents the data Flow in SDF unit for an 8-point ($n$) input. The butterfly is coupled with a delay of 4 (n/2) and a TWROM for twiddle factor storage. For the first four clock cycles, 0data path of the mux/demux is selected (shown with green lines) and coefficients are loaded into FIFO via $Fin1$ to provide a delay of $4$. The flow of coefficients in the FIFO register for each clock cycle is shown in Fig 13. For the next four clock cycles, $1$ data path is selected (shown with brown lines) and the coefficients from the $SDFin$ and the delay unit $Fout1$ arrive at $Bin1$ and $Bin2$ and are computed by the butterfly unit. As the inputs and outputs of the SDF unit are limited to one coefficient at a time, the first $4$ outputs from $Bout1$ are sent to $SDFout$ and outputs from $Bout2$ are sent back to the FIFO via $Fin2$ on each clock cycle, till $SDFout$ is occupied. After four clock cycles, $1$ data path is selected, and the data in FIFO is now sent to the $SDFout$. The mux/demux are

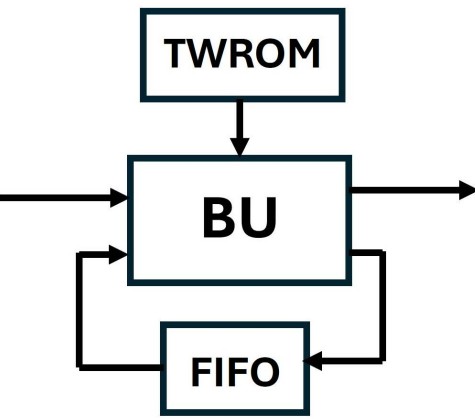

**Fig 11. SDF Unit.**

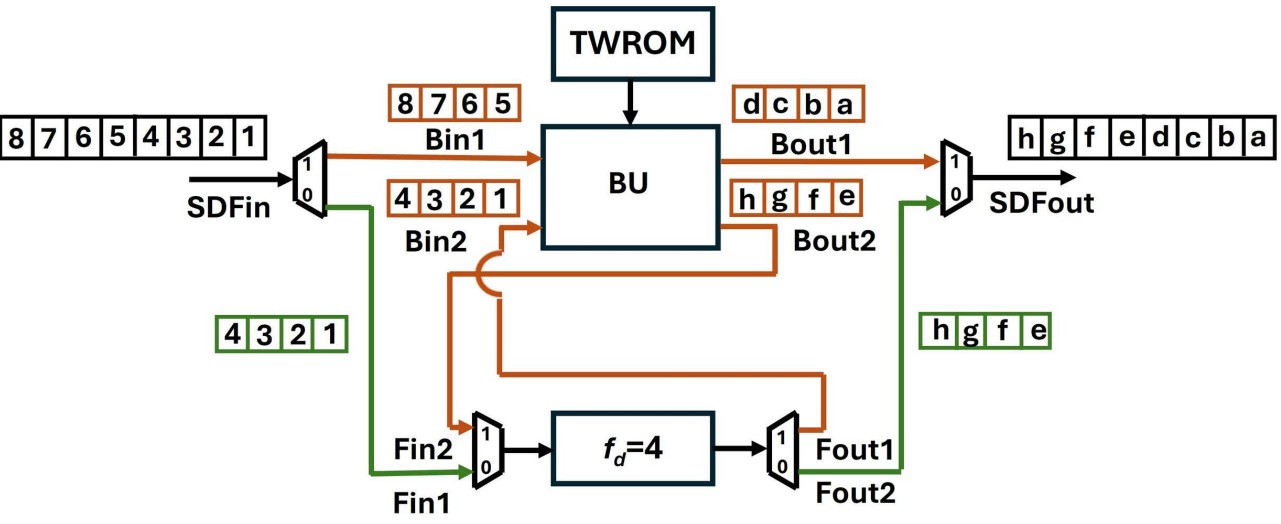

**Fig 12. Dataflow in an 8-point SDF unit.**

controlled by the *Muxsel* signal which has been generated in our design using counters. The TWROM, FIFO and counters in one stage are exclusively designed and independent of other stages.

**4.4.2. Kyber non-pipelined SDFNTT.** As $n$ is $256$ in Kyber and due to the truncated-NTT [38] used in Kyber explained in section 3.1, our SDFNTT consists of seven stages ($s = log_2 n - 1 = 7$). Seven SDF Processing units (PU) are cascaded as shown in Fig 14. Each PU consists of a radix-2 BU, TWROM and a delay unit.

The input and output of our Processing Unit (PU1) is one coefficient per cycle. Each PU represents one stage of NTT data flow. Dataflow for the first two stages is given in Fig 15. For the computation of radix-2 BU, coefficients at the input are required to differ with a specific offset for a particular stage. In the case of Kyber, input is a polynomial **p** of $256$ coefficients. These coefficients arrive at the input of PU1 per clock cycle and coefficients $\mathbf{p_1}, \mathbf{p_2}, \ldots, \mathbf{p_{128}}$ are loaded into the FIFO to provide a delay of $128$. The next set of coefficients $\mathbf{p_{129}}, \mathbf{p_{130}}, \ldots, \mathbf{p_{256}}$ are directed to the butterfly input $Bin1$ using the alternate mux data path, and coefficients $\mathbf{p_1}$ (output of FIFO) and $\mathbf{p_{129}}$ arrive at $Bin1$ and $Bin2$ at the same time. One output coefficient is sent as input to PU2, and the second coefficient is loaded again in the FIFO of PU1 to provide

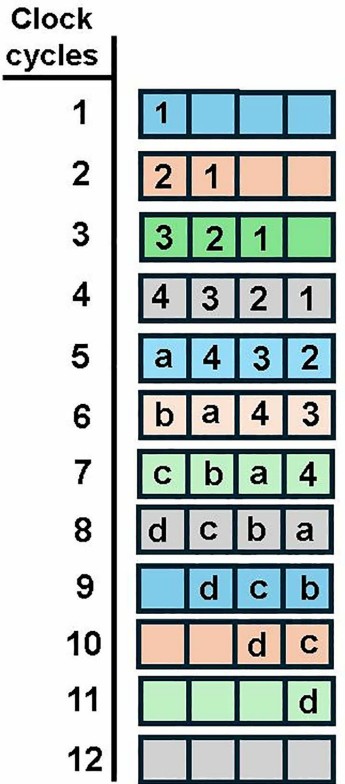

**Fig 13. Dataflow in FIFO for depth 4.**

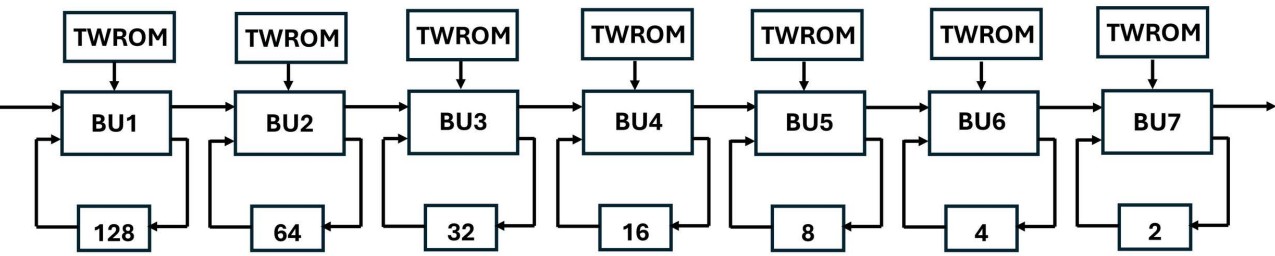

**Fig 14. PUs connected in Kyber SDFNTT.**

delay until the output path is available. Other input pairs $(\mathbf{p_2}, \mathbf{p_{130}})$, $(\mathbf{p_3}, \mathbf{p_{131}})$, .... separated by offset $128$ continue to arrive at the input at every clock cycle.

The next stage, PU2 processes the input $\mathbf{P}'$ in two sequences, first sequence comes from the output of BU of 1$^{st}$ stage $\mathbf{p}'_1, \mathbf{p}'_2, \ldots \mathbf{p}'_{128}$ and the next sequence comes from the FIFO $\mathbf{p}'_{129}, \mathbf{p}'_{130}, \ldots \mathbf{p}'_{256}$ of PU1. For the first 64 cycles, input coefficients $\mathbf{p}'_1, \mathbf{p}'_2, \ldots \mathbf{p}'_{64}$ goes to the FIFO and after $64$ cycles $\mathbf{p}'_{65}, \mathbf{p}'_{66}, \ldots, \mathbf{p}'_{128}$ are sent to *Bin*1. In this way, input coefficient pairs, $(\mathbf{p_1}, \mathbf{p_{65}})$, $(\mathbf{p_2}, \mathbf{p_{66}})$, ..., arrive at the butterfly unit with an offset of $64$. Subsequently, the second input sequence is computed by PU2, and the pattern repeats for the remaining stages.

**4.4.3. FIFO.** FIFO is used in the feedback for temporarily storing the input/output coefficients. The depth of FIFO for each stage $s$ is $f_d = n_s/2$, where $n_s$ is the size of the input sequence to the respective stage. For Kyber, the first

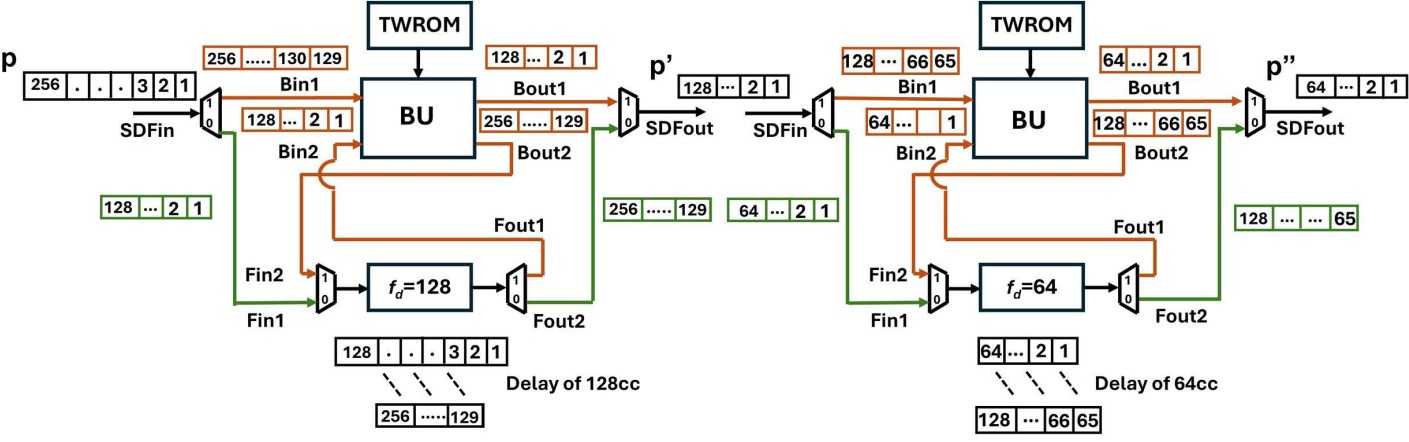

**Fig 15. Dataflow of first two stages for Kyber SDFNTT.**

stage has an input sequence of $256$ coefficients, hence $f_d = 128$. The input coefficient chunk for the second stage is $128$ coefficients so FIFO coupled with the second stage has a depth of $64$. This pattern continues for the rest of the stages.

**4.4.4. Twiddle factors storage and management.** An in-place NTT implementation stores the coefficients in BRAMs or registers and updates them directly at each stage of the butterfly computation. It implies, instead of reading from one memory array and writing results to another, the same memory location is reused. The proposed NTT design is based on SDF architecture, which differs from a classical in-place implementation. In our design, the polynomial coefficients are streamed through seven butterfly stages. Each stage uses dedicated FIFOs for memory handling and data alignment, and intermediate values are not stored in the same memory locations. Hence, this architecture does not follow the traditional in-place pattern but rather adopts an out-of place NTT approach with stage-wise dataflow for efficient pipelining and resource utilization. Moreover, our architecture is designed to be BRAM-free and instead uses FIFO buffers to stream polynomial coefficients. The design handles one polynomial at a time, and intermediate values are passed between the stages using dedicated FIFOs associated with each butterfly unit. This approach allows for a compact and efficient dataflow without the need for block RAM storage.

Twiddle factors are pre-computed values which are input to the BU unit during modular multiplication. These values can be stored in FPGA using embedded ROMs or LUT-based distributed ROMs. BROMs have limited width-depth configurations and will be under-utilized if used for storing the twiddle factors in our design, as polynomial in Kyber has $256$ coefficients of 12-bit each. The proposed architecture has seven stages, which makes instantiating seven BROMs for each stage costly in terms of hardware resources. LUT-based distributed memories are resource-friendly and can be placed anywhere near the rest of the logic in the FPGA design, also minimizing latency as the butterfly unit continuously loads data from the memory.

For NTT operation in Kyber, 128 (n/2) different twiddle factors are required. As INTT is symmetric, 128 (n/2) different twiddle factors are also needed during INTT operation. It means if unified NTT/INTT is designed (section 4.7) then Kyber of NTT must be able to store 256 (n) twiddle factors. The 1st stage of DIT NTT uses only the first value from the twiddle factor array to compute all butterfly operations. The 2nd stage uses second and third values and so on. For DIF NTT, the 1st stage uses the first $64$ values, the 2nd stage uses the next $32$ values, and the pattern continues for the remaining stages. This distribution of twiddle factors for different stages in NTT and INTT operation is shown in Fig 16, where blue boxes represent the number of twiddle factors for NTT operation and grey boxes represent twiddle factors for INTT operation.

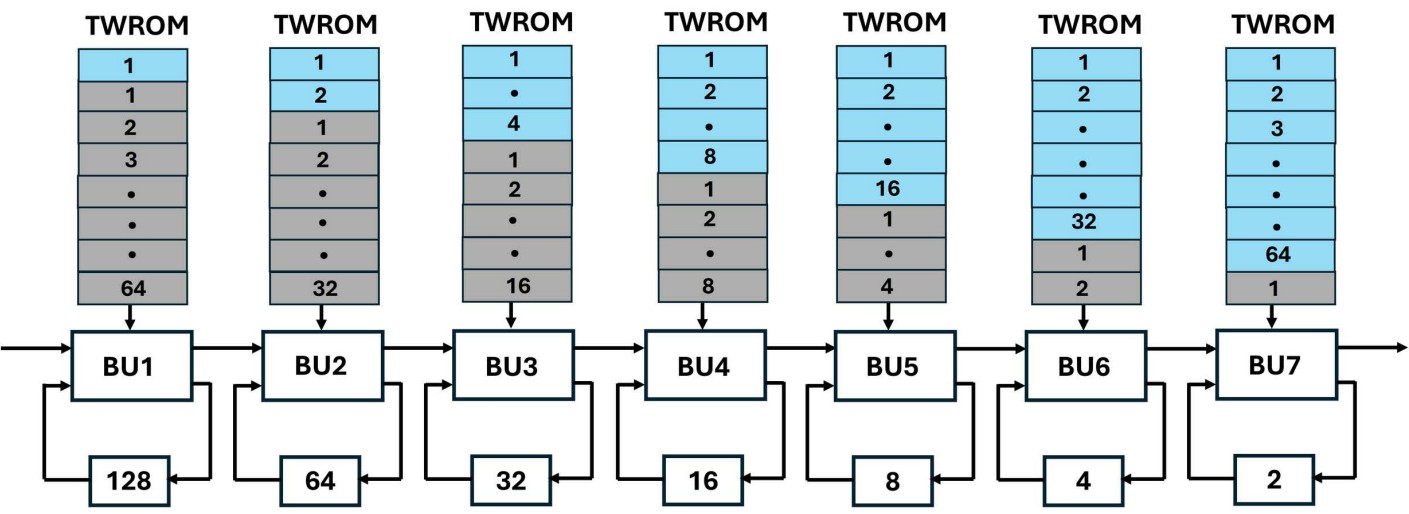

**Fig 16. Twiddle factors for NTT/INTT for different stages.**

In 7 series FPGA, 6 input LUTS are used to store up to $2^6 = 64$ bits and implement a $64 \times 1$ bit ROM [57]. Each stage in our SDFNTT has a different requirement of twiddle factors hence we have a different ROM capacity for all stages. For example, the first stage, when working in NTT mode needs 1 TW and when working in INTT mode requires 64 TW. The total number of twiddle factors for this stage is 65 and the width of each coefficient is 12 -bits, which are stored in a $65 \times 12$ bit ROM using 13 LUTS. The second stage requires 2 TW for NTT and 32 TW for INTT mode and are stored in a $34 \times 12$ bit ROM, implemented using 7 LUTS.

### 4.5. Data-conflict in pipelined SDF

Incorporating pipelining registers, increases the maximum frequency of the overall architecture. Data conflict arises when we adopt pipelining in conventional SDF architecture. The timing diagrams of using a non-pipelined and pipelined butterfly unit in SDFNTT are given in Figs 17 and 18 respectively. We have used a BU unit with a computation latency of 2 clock cycles (section 4.3) and discussed the data collision for an 8 -coefficient input sequence.

Coefficients $1, 2, 3, 4$ in Fig 18 are sent to the FIFO which provides a delay of 4 clock cycles to provide the required offset for butterfly computation between pairs $(1, 5)$, $(2, 6)$, $(3, 7)$, $(4, 8)$. The butterfly output $a, b, c, d$ goes to output while the coefficients $e, f, g, h$ are again sent to the FIFO until the output port is available. The second set of coefficients (shown by orange boxes), also continue to arrive at the input port. This causes data collision in the FIFO (yellow boxes). $C$ represents the collision in the FIFO between data output from $Bout2$ $(g, h)$ and second data input sequence $(1, 2)$. The delay between input and corresponding output coefficients is $6 = 4 + 2$ clock cycles, where 4 is the delay of the SDF architecture and 2 is the computational latency of the pipelined BU.

### 4.6. Pipelined SDFNTT

**4.6.1. Dataflow in pipelined SDF unit.** To provide a solution to the data collision in the FIFO register we have proposed SDF architecture with an alternate data flow which is configurable using multiplexers. Our new architecture of PU (Fig 19), employs four FIFO units, FIFO1 depth, $f1_d = n/4$, FIFO2 depth, $f2_d = n/4$, FIFO3 depth, $f3_d = (n/4) - 2$ and

| Clock | 1 | 2 | 3 | 4 | 5 | 6 | 7 | 8 | 9 | 10 | 11 | 12 | 13 | 14 | 15 | 16 | 17 | 18 | 19 | 20 | 21 |
|---|---|---|---|---|---|---|---|---|---|---|---|---|---|---|---|---|---|---|---|---|---|
| SDFin | 1 | 2 | 3 | 4 | 5 | 6 | 7 | 8 | 1 | 2 | 3 | 4 | 5 | 6 | 7 | 8 | | | | | |
| Fin | 1 | 2 | 3 | 4 | e | f | g | h | 1 | 2 | 3 | 4 | e | f | g | h | | | | | |
| Fout | | | | | 1 | 2 | 3 | 4 | e | f | g | h | 1 | 2 | 3 | 4 | e | f | g | h | |
| Bin1 | | | | | 5 | 6 | 7 | 8 | | | | | 5 | 6 | 7 | 8 | | | | | |
| Bin2 | | | | | 1 | 2 | 3 | 4 | | | | | 1 | 2 | 3 | 4 | | | | | |
| Bout1 | | | | | a | b | c | d | | | | | a | b | c | d | | | | | |
| Bout2 | | | | | e | f | g | h | | | | | a | b | c | d | | | | | |
| SDFout | ←— 4 cc —→ | | | | a | b | c | d | e | f | g | h | a | b | c | d | e | f | g | h | |

**Fig 17. Timing diagram of non-pipelined SDF (no data conflict).**

| Clock | 1 | 2 | 3 | 4 | 5 | 6 | 7 | 8 | 9 | 10 | 11 | 12 | 13 | 14 | 15 | 16 | 17 | 18 | 19 | 20 |
|---|---|---|---|---|---|---|---|---|---|---|---|---|---|---|---|---|---|---|---|---|
| SDFin | 1 | 2 | 3 | 4 | 5 | 6 | 7 | 8 | 1 | 2 | 3 | 4 | 5 | 6 | 7 | 8 | | | | |
| Fin | 1 | 2 | 3 | 4 | | | e | f | C | C | 3 | 4 | | | | | | | | |
| Fout | | | | | 1 | 2 | 3 | 4 | | | e | f | x | x | | | | | | |
| Bin1 | | | | | 5 | 6 | 7 | 8 | | | | | | | | | | | | |
| Bin2 | | | | | 1 | 2 | 3 | 4 | | | | | | | | | | | | |
| Bout1 | | | | | | | a | b | c | d | | | | | | | | | | |
| Bout2 | | | | | | | e | f | g | h | | | | | | | | | | |
| SDFout | ←— 6 cc —→ | | | | | | a | b | c | d | e | f | x | x | | | | | | |

**Fig 18. Timing diagram pipelined SDF (with data conflict).**

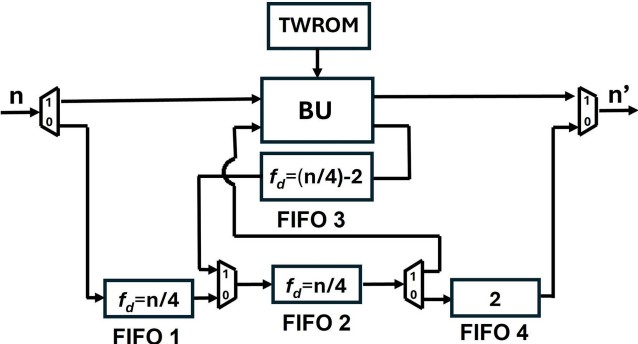

**Fig 19. Data collision free processing unit.**

FIFO4 depth, $f4_d = 2$. Different depths of the FIFOS in different paths are to control the data flow and to align the correct coefficient pairs at the input of the butterfly unit for computation.

Fig 20 shows the depth of FIFO registers for the first two stages of Kyber. For $1^{st}$ stage and input sequence $256$, $f1_d = 64$, $f2_d = 64$, $f3_d = 62$ and $f4_d = 2$. Similarly for $2^{nd}$ stage and input sequence $128$, $f1_d = 32$, $f2_d = 32$, $f3_d = 30$ and $f4_d = 2$. All seven PUs for Kyber NTT are connected and shown in Fig 21. The muxes select the data path and are configured by *Muxsel* signal, generated by counters in our design which are independent for each PU. Delay elements in our pipelined PU are $\{n/4, n/4, (n/4) - 2, 2\}$ unlike our previous architecture with the delay element of $n/2$. It implies that each PU of pipelined SDFNTT requires additional hardware of only $n/4$ delay unit and results in acceleration of the SDFNTT computation which is discussed in section 5.

The timing diagram of a $16$-coefficient input for pipelined SDFNTT providing a computation latency of $2$ cc is shown in Fig 22. The first half of input coefficients **1, 2, . . . , 8** are sent to the FIFO1 to provide a delay of $4$ clock cycles. The data is then directed to FIFO2 and delayed by another $4$ clock cycles. The output coefficients of FIFO2 are now aligned with the next half of the input coefficients **9, 10, . . . .., 16** arriving at *Bin*1 input. The butterfly unit gives output *B*1*out* **a, b, . . . , h** and *B*2*out* **i, j, . . . .p** after a latency of $2$ cc. *B*1*out* is directed to the output port, and *B*2*out* is delayed till the output port is occupied. We have designed the depths of FIFO $2$, $3$ and $4$ to control the data flow of *B*2*out* such that there is no collision in any of the FIFO units. At first, *B*2*out* is delayed by FIFO3 by $2$ cc. The output of FIFO3 is sent to FIFO2 which is now

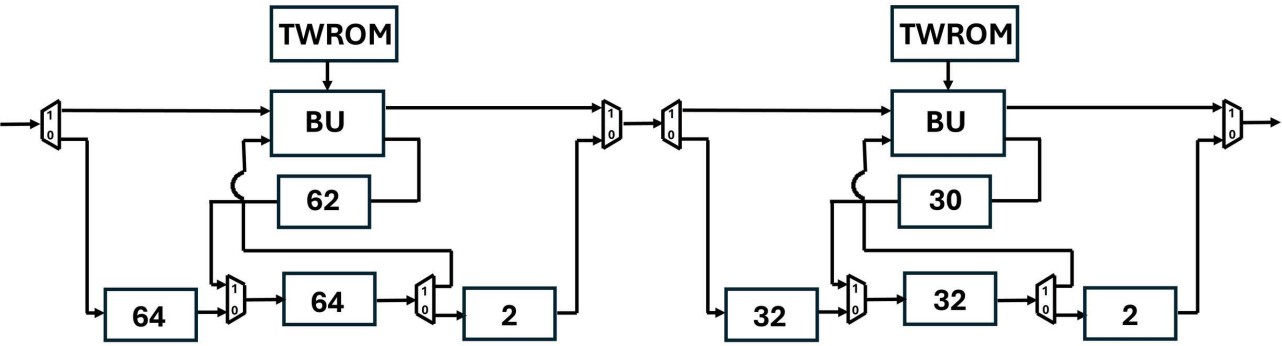

**Fig 20. First two stages of pipelined SDFNTT for Kyber.**

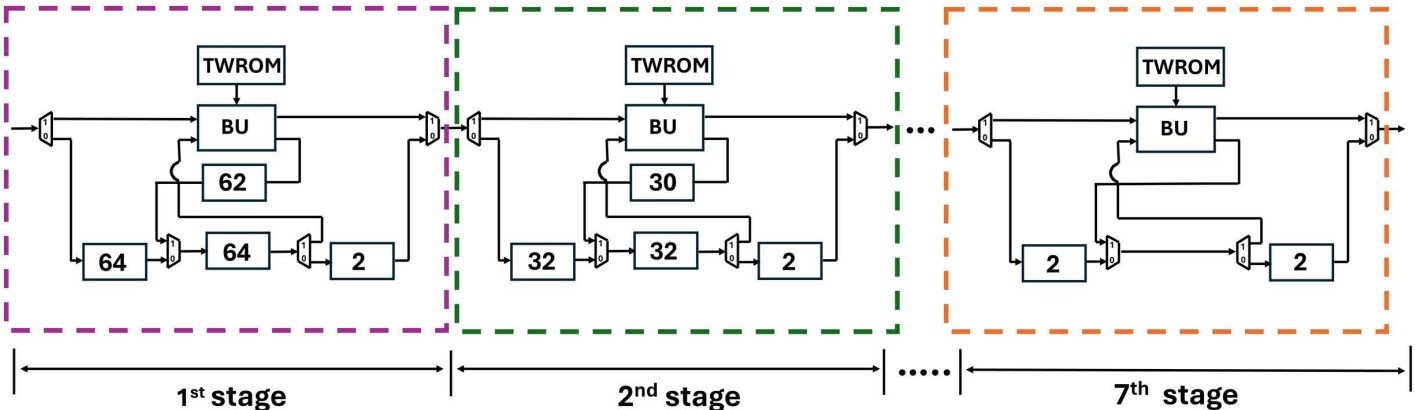

**Fig 21. PUs connected in pipelined SDFNTT for Kyber.**

| Clock | 1 | 2 | 3 | 4 | 5 | 6 | 7 | 8 | 9 | 10 | 11 | 12 | 13 | 14 | 15 | 16 | 17 | 18 | 19 | 20 | 21 | 22 | 23 | 24 | 25 | 26 | 27 | 28 | 30 | 31 | 32 | 33 | 34 | 35 |
|---|---|---|---|---|---|---|---|---|---|---|---|---|---|---|---|---|---|---|---|---|---|---|---|---|---|---|---|---|---|---|---|---|---|---|
| SDFin | 1 | 2 | 3 | 4 | 5 | 6 | 7 | 8 | 9 | 10 | 11 | 12 | 13 | 14 | 15 | 16 | 1 | 2 | 3 | 4 | 5 | 6 | 7 | 8 | 9 | 10 | 11 | 12 | 13 | 14 | 15 | 16 | | |
| F1in | 1 | 2 | 3 | 4 | 5 | 6 | 7 | 8 | | | | | | | | | 1 | 2 | 3 | 4 | 5 | 6 | 7 | 8 | | | | | | | | | | |
| F1out | | | | 1 | 2 | 3 | 4 | 5 | 6 | 7 | 8 | | | | | | | | | 1 | 2 | 3 | 4 | 5 | 6 | 7 | 8 | | | | | | | |
| F2in | | | | 1 | 2 | 3 | 4 | 5 | 6 | 7 | 8 | i | j | k | l | m | n | o | p | 1 | 2 | 3 | 4 | 5 | 6 | 7 | 8 | i | j | k | l | m | n | |
| F2out | | | | | | 1 | 2 | 3 | 4 | 5 | 6 | 7 | 8 | i | j | k | l | m | n | o | p | 1 | 2 | 3 | 4 | 5 | 6 | 7 | 8 | i | j | | | |
| B1in | | | | | | 9 | 10 | 11 | 12 | 13 | 14 | 15 | 16 | | | | | | | | | | | | | | | | | | | | | |
| B2in | | | | | | 1 | 2 | 3 | 4 | 5 | 6 | 7 | 8 | | | | | | | | | 1 | 2 | 3 | 4 | 5 | 6 | 7 | 8 | | | | | |
| B1out | | | | | | | | a | b | c | d | e | f | g | h | | | | | | | | | a | b | c | d | e | f | g | h | | | |
| B2out | | | | | | | | i | j | k | l | m | n | o | p | | | | | | | | | i | j | k | l | m | n | o | p | | | |
| F3in | | | | | | | | i | j | k | l | m | n | o | p | | | | | | | | | i | j | k | l | m | n | o | p | | | |
| F3out | | | | | | | | | | i | j | k | l | m | n | o | p | | | | | | | | | | | i | j | k | l | m | n | |
| F4in | | | | | | | | | | | | i | j | k | l | m | n | o | p | | | | | | | | | | | | | | i | j |
| F4out | | | | | | | | | | | | | | i | j | k | l | m | n | o | p | | | | | | | | | | | | | |
| SDFout | ◄ | | 10 cc | | | | ► | a | b | c | d | e | f | g | h | i | j | k | l | m | n | o | p | a | b | c | d | e | f | g | h | | | |

**Fig 22. Timing Diagram of Pipelined SDFNTT for 16 input coefficients.**

unoccupied and delays the coefficients by $4$ cc. Final delay of $2$ cc is provided by FIFO4. The output coefficients are now directed to the PU output port which is now available.

**4.6.2. Increasing pipelining.** The depth of FIFO3 and FIFO4 is proportional to the computation latency of the butterfly unit. If pipelining is increased by $3$, the depths of FIFOs are $f4_d = 3$ and $f3_d = (n/2) - 3$. If it is increased by $4$ then $f4_d = 4$ and $f3_d = (n/2) - 4$. This pattern can be continued to increase the number of pipelining but increasing pipelining also increases the delay of the overall architecture. Our analysis and tests conclude that the optimum tradeoff between architecture delay and $F_{max}$ can be achieved if BU latency is $2$ cc. Fig 23a and 23b present architectures of PU if BU latency is increased to 3 and $4$ respectively.

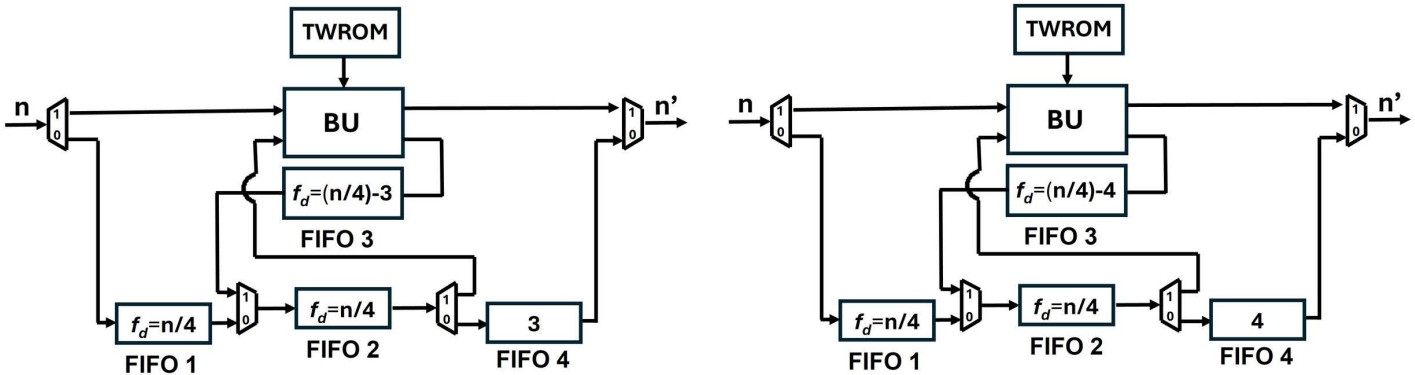

**Fig 23. Depth of FIFOs when pipelining of BU is (a) 3 cc (b) 4 cc.**

### 4.7. Unified SDF NTT/INTT architecture for Kyber

In section 4.3, reconfigurable butterfly unit have been proposed employing CT and GS butterfly structure. TWROM, storing the twiddle factor for seven stages of NTT and INTT operation has also been discussed in section 4.4.3. For designing a unified NTT and INTT architecture, we will adopt the resource sharing technique and use the same BU and TWROM unit in both computations. The only difference will be in the FIFO unit. Input order in DIT NTT and DIF INTT is different, so different depths of FIFO units are required. The unified NTT/INTT architecture for the first two stages of Kyber is shown in Fig 24. The brown mux/demux selects the path and the FIFO units configured in the PU unit. The selection signal for the mux/demux is provided by the *Mode* control signal, which is also given to the BUs for CT and GS mode selection. If *Mode* signal is $1$, CT structure is configured in BU and black data path is selected for the FIFO units. If *Mode* signal is $0$, BU becomes a GS butterfly structure for INTT operation and green data paths in PU are selected to provide delay according to the input sequence for INTT. As NTT and INTT are symmetric, the last stage of PU in NTT becomes the first stage of PU in INTT operation.

### 4.8. SDFNTT for Dilithium

Kyber and Dilithium are both lattice-based schemes, where Kyber is a key encapsulation mechanism (KEM) and Dilithium is a digital signature scheme. Both rely on the NTT for polynomial multiplication. The key difference lies in their modulus sizes: Kyber uses a 12-bit modulus ($q = 3329$), while Dilithium employs a 23-bit modulus ($q = 8380417$). This implies that Kyber requires narrower datapaths, whereas Dilithium demands significantly larger bit-widths when implemented in hardware.

The proposed SDF-based NTT architecture for Kyber can be adapted to support the Dilithium scheme by designing a butterfly architecture with wider bit-widths and integrating a modular multiplier for the larger modulus $q = 8380417$. This demonstrates the scalability of our architecture across different lattice-based cryptographic schemes. Reconfigurability and scalability of proposed SDFNTT can be achieved by developing a unified butterfly architecture capable of handling both $q = 3329$ and $q = 8380417$. Through such unified butterfly units, the proposed SDFNTT architectures can efficiently perform polynomial multiplications for both Kyber and Dilithium.

Kyber is not NTT-friendly for negative wrapped convolution (NWC), as its modulus $q = 3329$ does not support a primitive 2n-th root of unity. Therefore, Kyber adopts a truncated NTT approach, which requires only 7 stages of butterfly units for

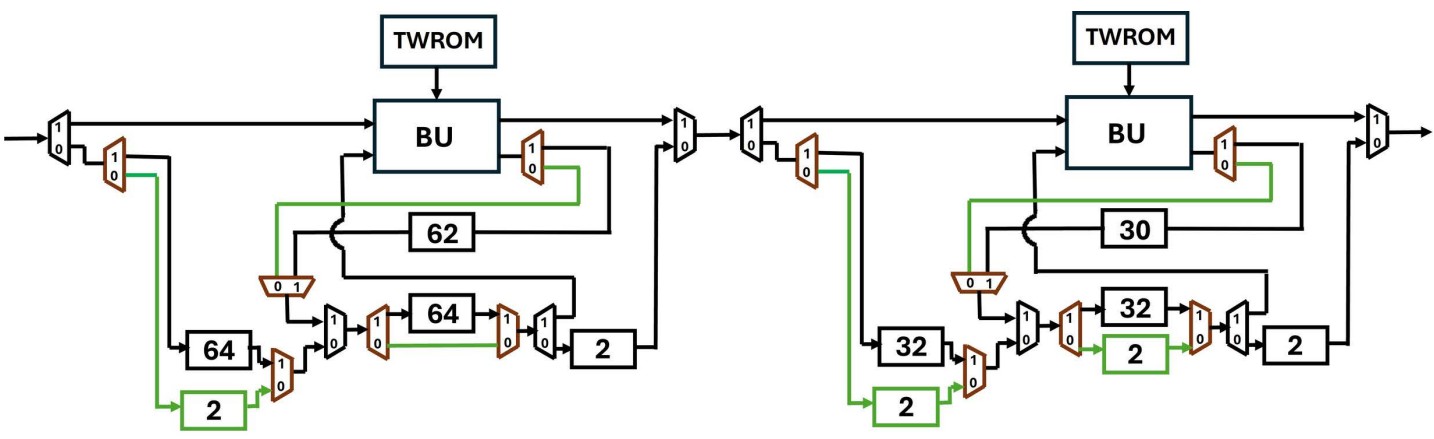

**Fig 24. First two stages of pipelined SDFNTT/INTT Architecture.**

n = 256. In contrast, Dilithium's modulus q = 8380417 is NTT-friendly, enabling a full NWC NTT. Consequently, a Dilithium SDFNTT implementation will require the complete 8 stages of butterfly units.

## 5. Result analysis

In this work we have developed two NTT architectures for CRYSTALS-Kyber. The comparison of our designs with state of the art pure NTT/INTT implementations in terms of consumed resources, latency, Area-Time Product (ATP) are shown in Table 2. Achieved ATP is also compared with prior works in Fig 25. We have implemented our designs using Verilog

**Table 2. Comparison of CRYSTALS-Kyber NTT/INTT to state-of-the-art designs.**

| Designs | Device | NTT/ INNT cycles | Operation | Freq (MHz) | Time (us) | #LUTs | #FFs | #DSPs | #BRAMs | ENS | AxT (ENS.ms) | TP/slice (Kb/slice) |
|---|---|---|---|---|---|---|---|---|---|---|---|---|
| [11] | Artix-7 | 446/446 | NTT/INTT | 237 | 1.88/- | 1401 | 2929 | 4 | 1 | 948.25 | 1.78 | 26.9 |
| [27] | Virtex-7 | 556/- | NTT/INTT | 235 | 1.2/- | 1362 | 1036 | 7 | 0.5 | 2212 | 2.6 | 4.6 |
| [29] | Artix-7 | 490/490 | NTT/INTT | 257 | 1.9/1.9 | 609 | 640 | 2 | 4 | 1137 | 2.16/2.16 | 22.1 |
| [28] | Artix-7 | 324/324 | NTT/INTT | 222 | 1.5/1.5 | 801 | 717 | 4 | 2 | 992.5 | 1.49/1.49 | 33.1 |
| [26] | Artix-7 | 902/902 | NTT/INTT | 208 | 4.34/4.34 | 274 | 181 | 1 | 1.5 | 462.5 | 2.0/2.0 | 11.9 |
| [58] | Virtex-7 | 1410/1540 | NTT/INTT | 290 | 4.86/5.31 | 1236 | 214 | 0 | 3 | 912 | 4.43/4.84 | 5.4 |
| [59]-a | Artix-7 | 896/1024 | NTT/INTT | 187 | 4.79/5.47 | 4731 | 3169 | 0 | 0 | 1193 | 5.7/6.5 | 4.2 |
| [59]-b | Virtex-7 | 896/1024 | NTT/INTT | 212 | 4.22/4.83 | 5109 | 3184 | 0 | 0 | 1287 | 5.4/6.2 | 4.4 |
| [22]-a | Artix-7 | 306/306 | NTT/INTT | 278 | 1.1/1.1 | 1070 | 1071 | 5 | 10.5 | 2825.5 | 3.1/3.1 | 15.4 |
| [22]-b | Virtex-7 | 306/306 | NTT/INTT | 296 | 1.03/1.03 | 982 | 1151 | 5 | 10.5 | 2887.8 | 2.97/2.97 | 16.1 |
| [60] | Airtex-7 | 2016 | NTT/INTT | 307.5 | 6.55 | 676 | 633 | 0 | 0 | 314 | 2.05/2.05 | 46.6 |
| **[a]** | **Artix-7** | **256/256** | **NTT/INTT** | **92** | **2.78/2.78** | **1703** | **333** | **0** | **0** | **426** | **1.18/1.18** | **10.1** |
| **[b]** | **Virtex-7** | **256/256** | **NTT/INTT** | **135** | **1.89/1.89** | **1703** | **333** | **0** | **0** | **426** | **0.8/0.8** | **14.8** |
| **[c]** | **Artix-7** | **270/270** | **NTT/INTT** | **153** | **1.76/1.76** | **1703** | **948** | **0** | **0** | **426** | **0.75/0.75** | **15.9** |
| **[d]** | **Virtex-7** | **270/270** | **NTT/INTT** | **198** | **1.36/1.36** | **1703** | **948** | **0** | **0** | **426** | **0.58/0.58** | **20.6** |

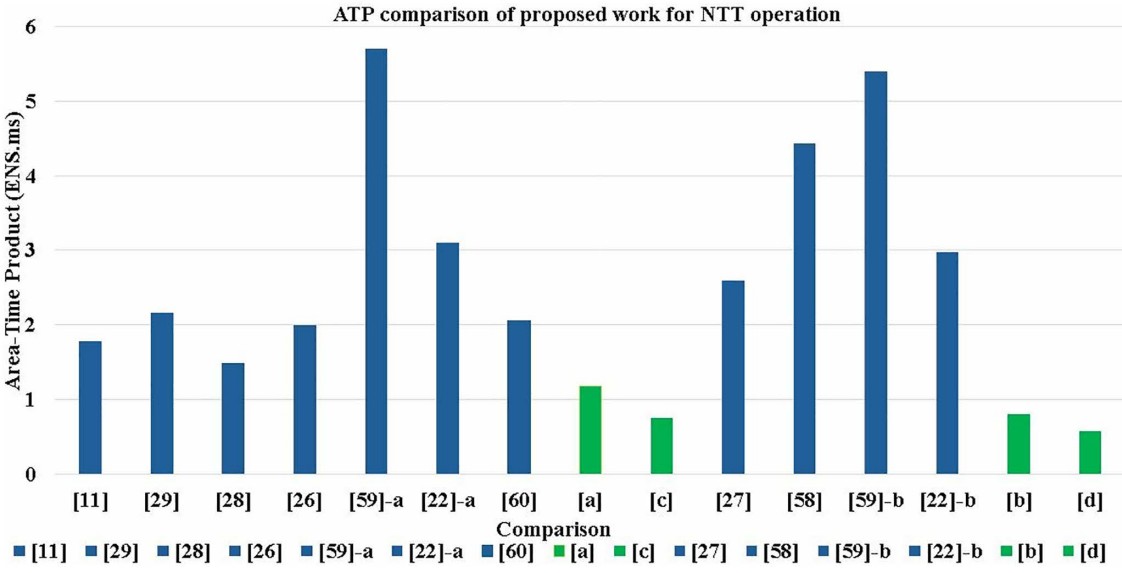

**Fig 25. Area-Time Product comparison of proposed designs a, b, c, d with previous work.**

on Artix-7 FPGA (XC7A100T-3) and Virtex-7 FPGA (XC7VX485T-3).Both synthesis and implementation were performed under the default Vivado settings. Each design was tested and verified through functional and post-place and route (PAR) simulations on Xilinx Vivado Design Suite 2022.2.

To provide fair comparisons, we normalized the LUT, DSP, and BRAM to the total area utilized in slices. The DSP is approximately equal to $100$ slices and $1$ BRAM is approximately equal to $196$ slices in Artix-7 according to [35]. In cases where ENS (Equivalent number of slices) is not reported, we have performed the conversion ourselves for comparison, as $1$ slice consists of $4$ LUTs and $8$ flip-flops (FFs) in 7-series FPGA [35]. Since some approaches focus on enhancing speed while others prioritize minimizing area, we have selected the ATP as a balanced metric to better represent overall performance. A smaller ATP value indicates a more efficient design. ATP is calculated by multiplying ENS to the execution time required to perform NTT/INTT operation.

### 5.1. NTT comparison

Table 2 indicates that our designs achieve the lowest ATP when compared to state-of-the-art designs. Low ATP of non-pipelined designs $(a, b)$ is attributed to low ENS utilized by employing resource sharing techniques and implementing a BRAM and DSP free architecture. In [11,22,26–29,58,59] DSP units are used for multiplication in modular multiplication unit and BRAMs are used for the data management and storing. As we have seven butterfly units in our design, utilizing seven BRAMs for storing pre-computed values and seven DSPs for modular multiplication would have significantly increased our resources. Our approach of using optimized and custom-built distributed LUT based BROMs for each stage and LUT based integer multiplier units resulted in a compact architecture.

The lowest ATP is achieved for pipelined designs $(c, d)$ due to inclusion of pipeline stages which reduces the critical path delay and results in an increase in the maximum frequency of the architecture. The increased frequency results in a decreased execution time, as Time (us)=Latency (cycles)/Frequency (MHz). The sequential and straightforward data flow of the input into the pipelined SDFNTT architecture, consequently leads to lowest NTT/INTT operation cycles when compared to [11,22,26–29,58,59], which employs a complex memory access pattern. These factors along with architectural optimization results in the lowest ATP for pipelined NTT/INTT designs.

Our first architecture is designed using conventional SDF architecture, without pipelining. Analysis of post-PAR utilization report indicates that our implementation takes $1703$ LUTs and $333$ FFs for Kyber achieving frequency of $92$ MHz on Airtex-7 and $135$ MHz on Virtex-7 devices. To improve the frequency of SDFNTT and to reduce the critical path delay, we added $2$ pipeline stages in our butterfly structure which resulted in a total of $14$ clock cycles latency in our design. Implementation of pipelined design utilizes $1703$ LUTs and $948$ FFs achieving a clock frequency of $153$ MHz on Airtex-7 and $198$ MHz on Virtex-7. Our pipelined designs achieve higher clock frequency, approximately $40\%$ on Airtex-7 and $32\%$ on Virtex-7 device when compared to the non-pipelined designs. Design $a, b, c$ and $d$ in Table 2 are non-pipelined (Artix-7), non-pipelined (Virtex-7), pipelined (Airtex-7) and pipelined (Virtex-7) implementations respectively.

Our non-pipelined SDFNTT $a, b$ takes $256$ clock cycles to compute the first output of NTT operation. The proposed pipelined SDFNTT $c, d$ employs a butterfly unit with computational latency of $2$ clock cycles. Seven pipelined butterfly units connected in cascade will result in latency of $14$ cc and delay of SDFNTT is $256 + 14 = 270$ clock cycles for both NTT and INTT operations. In [11] Nguyen et al. have presented an NTT/INTT architecture with $2 \times 2$ butterfly unit configuration. Our designs report $55\%$ reduction in ENS when compared to [11] and consequently implementation $a$ achieves $33.7\%$ and $c$ achieves $57.8\%$ improved ATP.

In [27] Ye et al. have implemented NTT core for various parameter sets. We have compared our work with design utilizing parameter set of Kyber ($n = 256, q = 3329$). When compared to proposed designs $b$ and $d$, resource utilization is reduced by $80\%$, which indicate our increased hardware efficiency. ATP is improved by $69.2\%$ when compared to $b$ and $77.7\%$ when compared to $d$. Zhang et al. in [29] have proposed a ping pong access scheme for memory management in iterative NTT/INTT design. In comparison with [29], our implementation achieves a $62.5\%$ reduction in resources

consumption and $45.3\%$, $65.3\%$ improved ATP when compared to *a* and *c*. K-RED based modular reduction algorithm and $2 \times 2$ butterfly configuration based NTT/INNT design is presented in [28] by Binesh et al. ENS in our implementation is reduced by $57.1\%$ and ATP in design *a* and *c* surpasses by $20.8\%$ and $49.6\%$ when compared to [28].

Itabashi in [26] has implemented radix-2, 2-parallel radix-2 and radix-4 architecture. We have compared our design with the radix-2 implementation comprising of one BU unit, two BRAMS and a TWROM unit. Results exhibit a $7.8\%$ reduction in ENS when [26] is compared to our implementation. ATP is also improved by $41\%$ and $62.5\%$ in designs *a* and *c*. Rashid et al. [58] have utilized extensive pipelining and resource re-use technique and implemented unified NTT/INTT core on Virtex-7 device. When comparing [58] to our designs, 53.2% reduction in resources is achieved along with 81.9%/83.5% and 86.9%/88% improved ATP for NTT/INTT operations when compared to *b* and *d*, respectively.

In [59], the authors have designed a unified NTT/INTT architecture by using register banks for memory storage and implemented their designs on both Artix-7 and Virtex-7 FPGA platforms. For comparison, we evaluate our designs *a* and *c* against [59]-a, while our designs *b* and *d* are compared with [59]-b. Our analysis shows 64% reduction in ENS and 79.3%/81.8%, 86.85%/88.5% improved ATP for NTT/INTT operations when *a* and *c* are compared to [59]-a. Designs *b* and *d* are 66.9% more resource efficient and achieves an improved ATP of 85.2%/87.09% and 89.2%/90.6% for NTT/INTT operations when compared to [59]-b.

In [22], Sun et al. have introduced a modified radix-4 butterfly unit and an enhanced K2-RED modular reduction for faster and more efficient NTT computation. The work is implemented on Aritex-7 [22]-a and Virtex-7 [22]-b FPGA devices. The resource efficiency is increased by 84.9% along with an ATP improvement of 61.9% and 75.8% when *a* and *c* are compared with [22]-a. Comparison of designs *b* and *d* with [22]-b reveals an 85.3% reduction in resources with a 73% and 80.5% increased ATP performance. The work in [60] presents three highly parallel designs on Artix-7 and uses interleaved multiplication for modular multiplication. We have compared the design with parameter $K = 2^2$ with our *a* and *c*. This results in 26.3% reduction in ENS and 42.4%/42.4%, 63.4%/63.4% improved ATP for NTT/INTT operations when *a* and *c* are compared to when [60].

Throughput per slice (TP/slice) values for NTT operation are also presented in Table 2. TP/slice is determined by first calculating throughput (TP) using {clock frequency x no. of bits}/latency. This is then divided by ENS to obtain TP/slice. A few designs listed in Table 2 report varying cycle counts for NTT and INTT operations. Since the execution cycles differ, the throughput for each operation also varies. It should be noted that TP and TP/slice were not directly reported in the referenced works but were instead calculated using the reported clock frequency, latency and the number of input bits processed per cycle in each design.

This section only compares TP/A of pipelined architectures (c, d) with the works listed in Table 2. By comparing the TP/A of c (Airtex-7) with prior designs, our work achieves a higher TP/A of 73.5%, 25%, 3.1% when compared to [59]-a, [22]-a, [26] and lower TP/A of 28%, 40.8%, 51.9% and 65.8% when compared to [11,28,29,60], respectively. Design d (Virtex-7) obtains a higher TP/A by 73.7%, 78.6%, 21.8% and 77.6%, when compared to [58,59]-b, [22]-b and [27], respectively.

While a few designs report higher TP/A, this comes at the cost of significantly increased area [11,28,29] or execution cycles [60]. In contrast, our approach enables efficient resource utilization and prioritizes overall system efficiency, achieving a better area-time product, which is especially beneficial for deployment on cost-sensitive or resource-constrained platforms. Furthermore, our serial architecture can be parallelized, offering a configurable trade-off between area and throughput.

## 5.2. Butterfly unit comparison

The low ENS in our designs is attributed to the use of a DSP-free optimized butterfly unit. A comparison of the resource utilization, ATP and throughput/slice of our butterfly unit with state-of-the-art designs from the literature is presented in Table 3. The butterfly unit in [32] uses a unified architecture and performs CT/GS butterfly operation for NTT/INTT

**Table 3. Comparison of Butterfly unit with state-of-the-art designs.**

| Design | LUT | FF | DSP | ENS | Freq (MHz) | Latency (CCs) | Time (ns) | ATP (ENS x us) | TP/slice (Mb/slice) |
|--------|-----|----|-----|-----|------------|---------------|-----------|----------------|---------------------|
| [32] | 312 | 207 | 1 | 178 | 192 | 3 | 15.6 | 2.78 | 8.6 |
| [26] | 274 | 181 | 1 | 168.5 | 208 | 5 | 24 | 4.05 | 5.9 |
| [61]-a | 185 | – | 1 | 146.3 | – | – | – | – | – |
| [61] -b | 152 | – | 1 | 138 | – | – | – | – | – |
| [23] | 289 | 236 | 0 | 72.3 | 256 | 7 | 27.3 | 1.9 | 12.1 |
| [62] | 107 | 64 | 1 | 126.8 | 280 | 3 | 10.7 | 1.35 | 17.6 |
| **[TW]** | **194** | **60** | **0** | **48.5** | **262** | **2** | **7.6** | **0.37** | **64.8** |

operation. Our DSP-free butterfly unit results in a 51.3% improved ATP and 86.7% higher throughput when compared with [32]. Three different configurations of butterfly units are presented in [26] to design NTT architectures. Proposed design in our work when compared with a single radix-2 butterfly unit in [26] achieves a 90.8% improved ATP and 90.8% improved throughput.

Authors in [61], have designed optimized butterfly unit for CRYSTALS-Kyber using different modular reduction techniques. Maximum operating frequency and latency is not given in [61], hence we have only compared the area utilization of our proposed butterfly with [61]. The butterfly units in [61] employing LUT4/KRED and $K^{1.5}$-red modular reduction units, achieve 66.8% and 64.8% greater area when compared to proposed butterfly. Nguyen et al. in [23] have proposed a pipelined butterfly unit to design a unified Kyber/Dilithium NTT architecture. The butterfly unit is DSP-free but the high latency results in an 80.5% increased ATP and 81.3% reduced TP/slice when compared to our proposed design. An optimized DSP-based butterfly unit is presented in [62]. The DSP-free optimized butterfly unit proposed in this paper achieves a 72.6% improved ATP and 72.8% high throughput when compared to [62].

## 5.3. Modular multiplier comparison

To achieve a compact and resource-efficient butterfly unit an area-time efficient modular multiplier is essential. In this work we have employed Montgomery reduction with lightweight operations, along with a LUT-based integer multiplier. A comparison of the ENS, ATP and throughput of our modular multiplier with state-of-the-art designs from the literature is presented in Table 4. Interleaved Multiplication approach is adopted in [63] for modular multiplication. The architecture is DSP-free but have high latency. Design [63]-a (PHIM approach) and [63]-b (PLIM approach) achieves 76.7%, 64.9% increased ATP and 76.7%, 65% lower throughput when compared to our modular multiplier.

A DSP-free modular multiplication is implemented in [64], but the Vedic multiplier and K-RED based-modular multiplier consumes increased LUTs/FFs then our implemented design. Our proposed modular multiplier achieves a 75% improved ATP and 75.2% increased throughput when compared to [64]. Optimized Barrett modular reduction and

**Table 4. Comparison of Modular Multiplier with state-of-the-art designs.**

| Design | LUT | FF | DSP | ENS | Freq (MHz) | Latency (CCs) | Time (ns) | ATP (ENSxus) | TP/slice (Mb/slice) |
|--------|-----|----|-----|-----|------------|---------------|-----------|--------------|---------------------|
| [63]-a PHIL | 99 | – | 0 | 38 | 391.8 | 12 | 30.6 | 1.16 | 10.3 |
| [63]-b PLIM | 111 | – | 0 | 41 | 372.2 | 7 | 18.8 | 0.77 | 15.5 |
| [64] | 228 | 174 | 0 | 77 | 283 | 4 | 14.2 | 1.08 | 11 |
| [62] | 60 | 56 | 1 | 115 | 320 | 3 | 9.4 | 1.07 | 11.13 |
| [61]-a | 83 | 38 | 1 | 120.8 | – | – | – | – | – |
| [61]-b | 56 | 0 | 1 | 114 | – | – | – | – | – |
| **[TW]** | **161** | **36** | **0** | **40.3** | **298** | **2** | **6.7** | **0.27** | **44.3** |

DSP-based integer multiplication approach is adopted in [62]. The DSP-free modular multiplier in this work achieves a 74.7% improved ATP and 74.8% higher throughput when compared to [62]. We have compared two modular reduction techniques in [61] with our work, LUT4/K-RED ( [61]-a) and K$^{1.5}$-RED ( [61]-b). LUT4/K-RED uses both lookup table and K-RED for reduction. K$^{1.5}$-RED is an improved variant of K-RED. Both designs perform integer multiplication using DSP resulting in high ENS. When compared to our proposed modular multiplier [61]-a and [61]-b achieves 66.6% and 64.6% increased ENS. ATP and TP/slice comparison is not possible as maximum frequency and clock cycles are not given in [61].

In [65], Shah et al presents PM (5x5 multiplier without reduction) and PFFM (5x5 multiplier with reduction) for post quantum digital signature scheme MAYO with q = 31. Our proposed multiplier is a 12 × 12 modular multiplier for CRYSTALS-Kyber (q = 3329). To facilitate a fair comparison of our core with [65], we re-instantiated our modular multiplier with a 5-bit operand and modulus q = 3329. Our 5x5 multiplier with reduction logic occupies 32 LUTs with CPD 2.17 ns as opposed to the 26 LUTs of PFFM in [65], with CPD 2.28 ns.

This modest increase is expected because [65] exploits the modulus q = 31, which allows a very lightweight reduction, whereas our design supports modulus q = 3329 and therefore incurs slightly higher logic cost even at smaller widths. Importantly, our design remains competitive in terms of CPD and it scales efficiently to the 12 × 12 setting required for Kyber. This comparison demonstrates the scalability and robustness of our architecture across different operand sizes.

## 5.4. Resource utilization analysis

Targeted Artix-7 platform has 63400 LUTS and 126800 FFs and Virtex-7 device has 303600 LUTs and 607200 FFs. Our design utilizes only 2.64% of the resources on the smallest Artix-7 device and 0.56% resource on Virtex-7. This exhibits that our lightweight NTT implementation is suitable to use in IOT constrained devices. The breakdown resource utilization of different modules in our implementation *c, d* is given in Figs 26 and 27. Moreover, pipelined SDFNTT consumes total

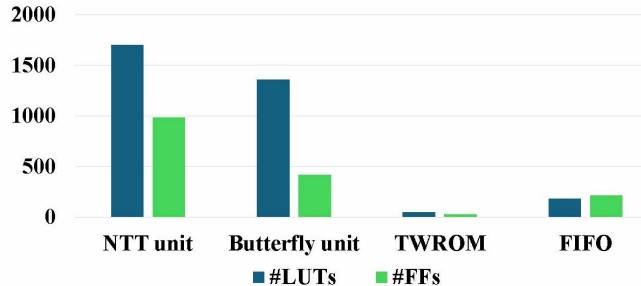

**Fig 26. Resource utilization and breakdown of implemented NTT/INTT core.**

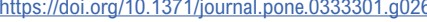

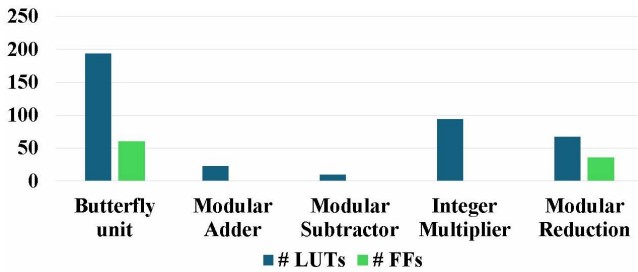

**Fig 27. Resource utilization and breakdown of implemented butterfly unit.**

on-chip power of 712 mW. Energy is obtained by multiplying the total power estimated in Vivado by the execution time of the design, which for design c is calculated as 1.25uJ.

## 6. Conclusion

CRYSTALS-Kyber has been standardized by NIST as public-key encryption and key encapsulation mechanism algorithm. In this research, we have accelerated the polynomial multiplication NTT unit of Kyber. BRAM-Free and DSP-Free, Single-path delay feedback (SDF) based NTT architecture for CRYSTALS-Kyber is presented. The performance of our hardware is attributed to adopting resource sharing technique, using distributed LUT-based BROMs for storing pre-computed values, LUT-based integer multiplier unit for coefficient multiplication and employing a compact multiplier-less reduction unit. Our proposed pipelined design achieves 49.6% better ATP and utilizes the least hardware resources when compared to former research in the literature.

Additionally, our architecture executes operations in constant time and therefore is resistant to timing-based side-channel attacks (SCAs). However, other types of side-channel leakages, such as power analysis or fault-injection-based attacks are possible in hardware designs. The current work did not incorporate dedicated countermeasures such as masking, hiding or noise injection as our primary focus was on achieving area–time efficient implementations but it is identified as future work and mentioned below.

## 7. Future work

The NTT is employed by CRYSTALS-Kyber to efficiently perform polynomial multiplication. In this paper, we implement a single-path delay feedback NTT architecture tailored for CRYSTALS-Kyber. To enhance the operating frequency of the overall design, pipelining is incorporated by addressing data conflict issues inherent in conventional SDF NTT structures. Our pipelined implementations, realized on Artix-7 and Virtex-7 FPGAs, demonstrate superior performance compared to state-of-the-art architectures, as detailed in Section 5. The designed SDF architecture can be extended to a multi-path data feedback (MDF) design, enabling the simultaneous processing of multiple NTT coefficients and increasing the overall throughput of the system. As a future direction, we aim to develop a unified butterfly architecture that supports both Kyber and Dilithium parameters while incorporating point-wise multiplication operations, thereby improving the scalability and adaptability of the design. Future work will focus on integrating countermeasures against power-based and fault-based SCAs to strengthen robustness while maintaining efficiency for lightweight PQC implementations.

## Author contributions

**Conceptualization:** Ayesha Waris.

**Data curation:** Ayesha Waris.

**Formal analysis:** Ayesha Waris.

**Investigation:** Ayesha Waris.

**Methodology:** Ayesha Waris.

**Project administration:** Arshad Aziz.

**Resources:** Ayesha Waris, Arshad Aziz.

**Software:** Ayesha Waris.

**Supervision:** Arshad Aziz, Bilal Muhammad Khan.

**Validation:** Ayesha Waris.

**Visualization:** Ayesha Waris.

**Writing – original draft:** Ayesha Waris.

**Writing – review & editing:** Arshad Aziz, Bilal Muhammad Khan.

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
