## [Decision Letter · Decision Letter 0]

14 Jul 2025

Dear Dr. Waris,

Thank you for submitting your manuscript to PLOS ONE. After careful consideration, we feel that it has merit but does not fully meet PLOS ONE’s publication criteria as it currently stands. Therefore, we invite you to submit a revised version of the manuscript that addresses the points raised during the review process.

We look forward to receiving your revised manuscript.

Kind regards,

Muhammad E. S. Elrabaa

Academic Editor

PLOS ONE

Journal Requirements:

Additional Editor Comments:

Please take into account all the reviewers' comments when revising the manuscript and prepare a detailed list of changes with the revised submission.

Reviewers' comments:

Reviewer's Responses to Questions

**Comments to the Author**

1. Is the manuscript technically sound, and do the data support the conclusions?

Reviewer #1: Yes

Reviewer #2: Yes

2. Has the statistical analysis been performed appropriately and rigorously?

Reviewer #1: Yes

Reviewer #2: Yes

3. Have the authors made all data underlying the findings in their manuscript fully available?

Reviewer #1: Yes

Reviewer #2: No

4. Is the manuscript presented in an intelligible fashion and written in standard English?

Reviewer #1: No

Reviewer #2: Yes

Reviewer #1: This article presents a pipelined and conflict-free NTT accelerator for CRYSTALS Kyber on an FPGA platform. It employed several architectural optimizations, including pipelining, resource sharing, and algorithmic optimizations like the multiplier-less Montgomery reduction algorithm. However, the following concerns need to be addressed:

1. Table II is not complete; add results for throughput per unit area, like throughput per slice. Also comment on the power consumption of the design.

2. References are used in a very strange manner, like (1) and (2). This representation is used for equations, not for references.

3. A lot of typos and grammatical mistakes; it is advised to proofread the manuscript very carefully. In addition, a few lines of paragraphs are not recommended, so reduce the number of paragraphs by increasing their sizes.

4. The authors must explain whether it is in-place NTT or any other form. How many BRAMs are used to store the polynomial coefficients, and is it for a single polynomial?

5. The authors didn’t mention their previous work, ‘Area-time efficient pipelined number theoretic transform for CRYSTALS-Kyber’, is there any good reason?

6. Provide your modular multiplier, butterfly unit results, and compare them with: (1) ‘efficient reconfigurable modular multipliers for post-quantum digital signatures’, and (2) ‘efficient soft-core multipliers for digital signatures’. Add this comparison to the manuscript.

7. Some recent work must be added and compared in Table 2, such as (1) Efficient Number Theoretic Transform Architecture for CRYSTALS-Kyber, etc.

8. Most of the schemes are now hybrid, so how will your design scale in a multi-scheme environment where lattice and hash-based schemes are integrated?

9. Power consumption is very important; add power consumption stats and compare them with the listed designs.

10. Side-channel attacks are very important nowadays. How robust is your design against these attacks (power and timing)?

Reviewer #2: This paper presents a new pipelined hardware implementation of NTT for Kyber on FPGAs which provides significant improvement in ATP.

I have to following comments to improve the quality of the manuscript.

1. When a new implementation is proposed it must be analyzed from side-channel aspects. It is suggested that the authors explain the potential side-channel leakages of their architecture (e.g., timing, power)?

If there are any, do they have considered any potential countermeasures (e.g., constant-time logic, masking, etc.)?

2. It is beneficial to add a discussion and explain the challenges of adopting the proposed approach to other LWE schemes that use NTT such as Dilithium.

3. Please also add power and energy metrics of your design.

4. Please mention the synthesis and implementation settings that you used on Vivado.

Also, there are several typos and errors throughout the manuscript. Please proofread the whole manuscript several times. The following are some of these issues I noticed.

5. Picture qualities are bad especially when zoomed. It is suggested that the authors import the pdf file of their figures instead of jpeg, png, etc.

6. Eq 1??

7. CRYSTALs must be CRYSTALS

8. (8),when (no space)

9. [52]

10. Analysis show(s)

11. An acronym must be defined only once when it is used for the first time. From there it must be redefined again. Do this to all the acronyms you use, including ENS.

Overall, I believe the paper has merits, but it requires substantial edits before being ready for publication.

**Do you want your identity to be public for this peer review?** For information about this choice, including consent withdrawal, please see our Privacy Policy

Reviewer #1: No

Reviewer #2: No

---

## [Author Response · Author response to Decision Letter 1]

29 Aug 2025

The Authors are thankful to the Reviewers for their suggestions and have incorporated all the points raised during the review process.

---

## [Decision Letter · Decision Letter 1]

11 Sep 2025

Pipelined and Conflict-Free Number Theoretic Transform Accelerator for CRYSTALS-Kyber on FPGA

PONE-D-25-24864R1

Dear Dr. Waris,

We’re pleased to inform you that your manuscript has been judged scientifically suitable for publication and will be formally accepted for publication once it meets all outstanding technical requirements.

Kind regards,

Muhammad E. S. Elrabaa

Academic Editor

PLOS ONE

Additional Editor Comments (optional):

Reviewer #1:

Reviewer #2:

Reviewers' comments:

Reviewer's Responses to Questions

**Comments to the Author**

Reviewer #1: All comments have been addressed

Reviewer #2: All comments have been addressed

2. Is the manuscript technically sound, and do the data support the conclusions?

Reviewer #1: Yes

Reviewer #2: Yes

3. Has the statistical analysis been performed appropriately and rigorously?

Reviewer #1: Yes

Reviewer #2: Yes

4. Have the authors made all data underlying the findings in their manuscript fully available?

Reviewer #1: Yes

Reviewer #2: No

5. Is the manuscript presented in an intelligible fashion and written in standard English?

Reviewer #1: Yes

Reviewer #2: Yes

Reviewer #1: (No Response)

Reviewer #2: I have no more comments. But it is highly encouraged that the authors publish their codes to be used by others.

**Do you want your identity to be public for this peer review?** For information about this choice, including consent withdrawal, please see our Privacy Policy

Reviewer #1: No

Reviewer #2: No

---

## [Editor Report · Acceptance letter]

PONE-D-25-24864R1

PLOS ONE

Dear Dr. Waris,

I'm pleased to inform you that your manuscript has been deemed suitable for publication in PLOS ONE. Congratulations! Your manuscript is now being handed over to our production team.

Kind regards,

on behalf of

Dr. Muhammad E. S. Elrabaa

Academic Editor

PLOS ONE